# DE NOVO PROTEIN DESIGN USING GEOMETRIC VECTOR FIELD NETWORKS

**Weian Mao**[1,2]*, **Muzhi Zhu**[1]*, **Zheng Sun**[3]*, **Shuaike Shen**[1],
**Lin Yuanbo Wu**[3], **Hao Chen**[1], **Chunhua Shen**[1,4]

[1] Zhejiang University, China    [2] The University of Adelaide, Australia
[3] Swansea University, UK    [4] Ant Group

## ABSTRACT

Innovations like protein diffusion have enabled significant progress in *de novo* protein design, which is a vital topic in life science. These methods typically depend on protein structure encoders to model residue backbone frames, where atoms do not exist. Most prior encoders rely on atom-wise features, such as angles and distances between atoms, which are not available in this context. Thus far, only several simple encoders, such as IPA (Jumper et al., 2021), have been proposed for this scenario, exposing the frame modeling as a bottleneck. In this work, we proffer the Vector Field Network (VFN), which enables network layers to perform learnable vector computations between coordinates of frame-anchored virtual atoms, thus achieving a higher capability for modeling frames. The vector computation operates in a manner similar to a linear layer, with each input channel receiving 3D virtual atom coordinates instead of scalar values. The multiple feature vectors output by the vector computation are then used to update the residue representations and virtual atom coordinates via attention aggregation. Remarkably, VFN also excels in modeling both frames and atoms, as the real atoms can be treated as the virtual atoms for modeling, positioning VFN as a potential *universal encoder*. In protein diffusion (frame modeling), VFN exhibits an impressive performance advantage over IPA, excelling in terms of both designability (**67.04**% vs. 53.58%) and diversity (**66.54**% vs. 51.98%). In inverse folding (frame and atom modeling), VFN outperforms the previous SoTA model, PiFold (**54.7**% vs. 51.66%), on sequence recovery rate. We also propose a method of equipping VFN with the ESM model (Lin et al., 2023), which significantly surpasses the previous ESM-based best result (**62.67**% vs. 55.65%), LM-Design (Zheng et al., 2023), by a substantial margin. Code is available at https://github.com/aim-uofa/VFN

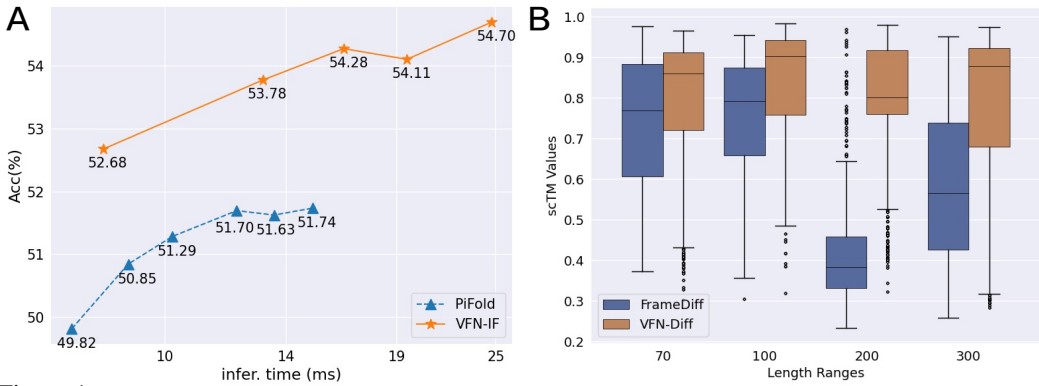

Figure 1: Experimental results on protein diffusion and inverse folding. A) VFN-IF compared to PiFold with varying numbers of layers, showcasing the trade-off between speed and sequence recovery rate. B) VFN-Diff compared to FrameDiff (IPA) for designability across proteins of varying lengths.

---

*WM, ZS and MZ contributed equally. Work was done when WM was visiting Zhejiang University. HC and CS are the corresponding authors.

# 1  INTRODUCTION

The field of *de novo* protein design (Huang et al., 2016) represents a pivotal frontier in the realms of bioengineering and drug development, holding the potential to bring about a revolutionary shift in the creation of innovative therapeutic agents. Recent advancements in this domain have been driven by a groundbreaking paradigm that combines protein structure diffusion models (Watson et al., 2023; Yim et al., 2023) with inverse folding networks (Dauparas et al., 2022). Specifically, this paradigm initiates by employing a protein diffusion model to stochastically generate the backbone structure of a protein, represented by residue frames. Since the types of amino acids in the generated protein are initially unknown, an inverse folding network is then utilized to design the protein sequences for each residue based on the backbone residue frames. While this paradigm has shown great success for protein design, it also brings challenges for deep learning-based protein structure encoders.

Methods like protein diffusion rely on protein structure encoders to model and sample protein structures. However, prior structure encoders are unsuitable or severely limited in this case, primarily because the representation of residues is highly specialized in methods like diffusion. Specifically, in those methods, backbone frames are commonly employed to represent the spatial information of residues, with atom-level representations typically absent. The absence of atoms has rendered most previous encoders ineffective for protein design, as they typically rely on atom-level input features such as interatomic angles and distances. Although a few basic encoders, such as IPA (Jumper et al., 2021), were designed for frame modeling, they still faced significant limitations. For instance, IPA simply performs distance pooling between frame-anchored virtual atoms as geometric features. Obviously, such pooling operation and the single pooled distance value clearly lacks the expressiveness required for representation. We refer to these limitations as the *atom representation bottleneck*, which is explained in detail in §2.4. In response to this challenge, we propose a novel structure encoder called the Vector Field Network (VFN).

The core idea of VFN revolves around the utilization of a vector-specific linear layer to extract multiple geometric feature vectors by mapping the coordinates of frame-anchored virtual atoms. Specifically, VFN introduces virtual atoms in Euclidean space for each amino acid, and these virtual atoms move in conjunction with the frames, serving as dynamic representations of the frames. When modeling the relationship between two amino acids, a module called the 'vector field operator' takes the virtual atom coordinates of the two amino acids as vector inputs and performs operations similar to a linear layer. Within the vector field operator, each vector is initially multiplied by a learnable scalar weight, and these weighted vectors are accumulated to compute the output feature vector. Like a linear layer, the vector field operator has multiple output channels, with each output channel generating a Euclidean vector. Subsequently, these output feature vectors are directed into an MLP layer for processing, facilitating the fusion of residue features and enabling the modeling of residue frames.

VFN exhibits significant advantages over previous encoders in terms of its expressive power and versatility, as VFN circumvents the *atom representation bottleneck* present in IPA. As previously mentioned, IPA relies on a single scalar variable, which represents the sum of distances between virtual atoms, serving as both geometric features and attention bias. This single scalar variable and corresponding pooling operation limit the expressive capacity of IPA. In contrast, VFN can flexibly extract multiple feature vectors through a vector field operator, thereby circumventing this bottleneck. Please refer to §2.4 for more details.

In scenarios where frame and backbone atoms coexist, such as inverse folding, VFN maintains excellent expressiveness while simultaneously offering enhanced generality and flexibility. This is because real atoms can be treated as virtual atoms in VFN, and the coordinates of both real and virtual atoms can be used to model the relationships between amino acids using the vector field operator, naturally forming hierarchical representations. Compared to other atom-based inverse folding methods, VFN achieves superior performance by potentially modeling frames more effectively. Thus, VFN also surpasses the current SoTA in inverse folding. It is worth emphasizing that, IPA is not suitable for treating the real atoms as the virtual atoms, since the IPA network cannot provide learnable weights for the coordinates of each atom, facing the *atom representation bottleneck*.

To assess the performance of VFN for *de novo* protein design, we have implemented two models based on VFN, namely VFN-IF and VFN-Diff, tailored for protein inverse folding and diffusion based generative modeling, respectively. Experimental results consistently demonstrate the remarkable

performance of VFN. For protein diffusion, VFN-Diff significantly outperforms the prior solid baseline, FrameDiff (Yim et al., 2023), in terms of designability (**67.04**% *vs.* 53.58%) and diversity (**66.54**% *vs.* 51.98%). It is important to emphasize that the key distinction between VFN-Diff and FrameDiff lies in the replacement of the encoder—IPA, with VFN layers. This substantiates the superior geometric feature extraction capability of VFN over the widely adopted IPA.

In the domain of inverse folding, VFN-IF exhibits a substantial performance boost over the current state-of-the-art method, PiFold (**54.74**% *vs.* 51.66%, sequence recovery rate). Furthermore, we have trained a larger-scale VFN-IF model on the entire PDB database, achieving an impressive sequence recovery rate of 57.14%, underscoring the scalability of VFN-IF. Additionally, we propose a novel variant of VFN-IF, called VFN-IFE, which is equipped with an external knowledge base, achieves remarkable precision at **62.64**%, surpassing SoTA approaches in this regard, LM-Design (Zheng et al., 2023) (55.65%), by a substantial margin.

The main contributions of this work can be summarized as follows:

- We propose the Vector Field Network (VFN), which employs a vector field operator to extract geometric feature vectors between frame-anchored virtual atoms, resembling a linear layer. This approach overcomes the *atom representation bottleneck*, thereby enhancing representational capabilities. Notably, VFN can also incorporate real atoms as the virtual atoms for hierarchical modeling, positioning it as a potential *universal encoder*.

- For protein diffusion, VFN significantly enhances the designability(**67.04**% *vs.* 53.58%) and diversity(**66.54**% *vs.* 51.98%) of protein generation compared to IPA.

- For inverse folding, VFN significantly surpasses the previous SoTA model, PiFold (**54.74**% *vs.* 51.66%, sequence recovery rate). Additionally, we propose a method for equipping VFN with an external knowledge base, achieving a substantial breakthrough over the SoTA approach, LM-Design, in this regard, thus elevating model accuracy to next level (**62.64**% *vs.* 55.65%).

## 2 RELATED WORK

### 2.1 DE NOVO PROTEIN DESIGN

*De novo* protein design, which involves the creation of proteins from scratch, holds paramount significance in the fields of enzyme engineering and protein engineering. Traditional approaches such as RosettaDesign (Liu & Kuhlman, 2006) were prevalent before the advent of machine learning-based methods. In recent years, with the maturation of machine learning techniques, advanced deep learning-based methods have emerged, exemplified by Huang et al. (2022) (Side Chain Unknown Backbone Arrangement) and protein diffusion (Watson et al., 2023; Fu et al., 2023).

The paradigm of protein diffusion is regarded as one of the most promising methods in the realm of protein design, which encompasses protein diffusion (Yim et al., 2023) and inverse folding (Gao et al., 2022; Jendrusch et al., 2021; Wu et al., 2021; Ovchinnikov & Huang, 2021; Dauparas et al., 2022; Ingraham et al., 2019; Hsu et al., 2022; Gao et al., 2024; Derevyanko et al., 2018). Specifically, a protein diffusion model first generates the backbone structure of a protein, followed by an inverse folding network that designs the corresponding sequence for this backbone. The feasibility of both these steps has been experimentally validated through cryo-electron microscopy (Watson et al., 2023; Dauparas et al., 2022), marking a significant breakthrough in the field of protein design. However, while protein diffusion methods based on frame representation achieve significant success, in these methods, atom representation is absent, rendering previous general purpose encoders unusable.

### 2.2 GENERAL PURPOSE ENCODER

In the past, numerous encoders (Hermosilla et al., 2020; Zhang et al., 2022; Hermosilla & Ropinski, 2022; Veličković et al., 2018; Baldassarre et al., 2021; Li et al., 2022; Shroff et al., 2019; Dumortier et al., 2022; McPartlon et al., 2022; Cao et al., 2021; Anishchenko et al., 2021; Karimi et al., 2020; Zhang et al., 2020; Wang et al., 2022b; Derevyanko et al., 2018) have been proposed for tasks such as model quality assessment (Townshend et al., 2021) and fold classification (Hou et al., 2018), where atomic information is available. However, these methods are not suitable for protein design tasks where atomic representations of proteins are unavailable. For instance, GVP (Jing et al., 2020)

transforms input atomic coordinates into vector and scalars variables as the network input, facilitating the model's SE(3) invariance. Meanwhile, Wang et al. (2022a); Jin et al. (2022) proposed efficient approaches for modeling protein structures using hierarchical protein representations, enabling GNNs to possess both residue-level and atom-level hierarchical representations. Nevertheless, such approaches are evidently less applicable in the methods like protein diffusion, as atomic information is unattainable.

## 2.3 FRAME-BASED ENCODER

The use of frame-based representations in protein structure encoders has been relatively unexplored, with few existing encoders being rather rudimentary in nature. Historically, these encoders have primarily served as auxiliary modules in protein structure prediction models, such as RoseTTAFold (Baek et al., 2021) and AlphaFold2 (Jumper et al., 2021). In RoseTTAFold, the explicit concept of frames is absent; instead, it employs the origin of frames to represent frames and employs SE(3)-transformers (Fuchs et al., 2020) to model these frames. However, this approach conspicuously neglects rotational information of frames, considering only positional offsets. AlphaFold2, on the other hand, introduces the IPA structural encoding module to address this limitation. IPA represents frames using framed-anchored virtual atoms and extracts geometric information between two frames by computing the sum of distances between virtual atom pairs. This approach takes into account both the coordinate offsets between frames and their rotational information.

## 2.4 ATOM REPRESENTATION BOTTLENECK

However, IPA still faces the *atom representation bottleneck*. Firstly, IPA does not incorporate learnable weights when extracting features for virtual atoms; it directly applies a simple summation pooling operation to the atom-pair distances, thereby lacking flexibility. Secondly, the sum of distances between virtual atom pairs yields only a scalar variable, making it challenging to represent complex geometric information, thus lacking expressiveness. In VFN, these issues have been successfully resolved. VFN introduces an operator similar to a linear layer, characterized by the inclusion of learnable weights associated with individual atom coordinates. These weights facilitate flexible vector computations, effectively mitigating the flexibility challenges inherent in IPA's pooling operation. Furthermore, the operator in VFN can return multiple feature vectors, eliminating the expressiveness bottleneck caused by the single scalar variable (the pooled distance) in the IPA. For further details regarding the vector field operator, please refer to §3.2.

## 3 VECTOR FIELD NETWORK LAYERS

The Vector Field Network (VFN) is designed to extract geometric features between amino acids using a module, named the vector field operator (refer to §3.2). In each layer of VFN, the protein's representation undergoes a sequential process including the vector field operator, node interactions (§3.3), edge interactions (§3.4) and virtual atom updating (§3.4). The vector field operator is crucial for extracting geometric features between pairs of amino acids through vector computations and virtual atoms. Subsequently, these extracted geometric features are aggregated and employed to update the representation of each amino acid through node and edge interactions. At the end of each layer, the coordinates of the mentioned virtual atom are updated by aggregation or residue representations. The mentioned modules and the overall pipeline are elucidated subsequently.

## 3.1 PROTEIN REPRESENTATION

In VFN, a protein consisting of $n$ amino acids is represented as a graph denoted as $\mathcal{G} = (\mathcal{S}, \mathcal{E}, \mathcal{T}, \mathcal{Q})$. Here, $\mathcal{S} = \{\mathbf{s}_i \in \mathbb{R}^{d_v}\}_{i=1,\ldots,n}$ represents the set of node features. To encode the positional information of each amino acid in space, we use a set of local frames, $\mathcal{T} = \{\mathbf{T}_i\}_{i=1,\ldots,n}$, to represent the position of each amino acid. We introduce a set of frame-anchored virtual atoms, whose coordinates are maintained by Eq. equation 8 or Eq. equation 9. Specifically, we denote all virtual atom coordinates as $\mathcal{Q} = \{\mathbf{Q}_i\}_{i=1,\ldots,n}$, with $\mathbf{Q}_i = \{\vec{\mathbf{q}}_k \in \mathbb{R}^3\}_{k=1,\ldots,d_q}$ representing a set of virtual atom coordinates associated with the $i$-th residue *w.r.t.* $\mathbf{T}_i$. $d_q$ denotes the number of virtual atoms in each residue. These virtual atom coordinates are treated as vectors in the vector field operator, allowing for vector computations to extract geometric features. Additionally, we construct edges in the graph based on specific rules, such as a complete graph or $k$-nearest neighbors. The complete graph is taken by default, so the set of edge features can be denoted as $\mathcal{E} = \{\mathbf{e}_{i,j} \in \mathbb{R}^{d_e}\}_{i,j=1,\ldots,n}$.

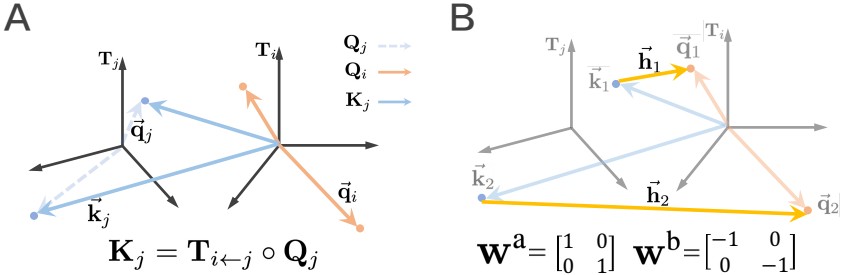

Figure 2: Pipeline for the Vector Field Operator. A) Transforming the virtual atomic coordinates $\mathbf{Q}_j$ from frame $\mathbf{T}_j$ to frame $\mathbf{T}_i$ to obtain $\mathbf{K}_j$. B) An example of vector computation involving vectors $\mathbf{Q}_i$ and $\mathbf{K}_j$ using learnable weights $\mathbf{w}^a$ and $\mathbf{w}^b$ as defined in Equation equation 2. When $\mathbf{w}^a$ and $\mathbf{w}^b$ are specific weights (as shown in figure), the vector field can yield the Euclidean vector, $\vec{\mathbf{h}}_1$ and $\vec{\mathbf{h}}_2$, between two particular atoms.

## 3.2 Vector Field Operator

As shown in Figure 2, the vector field operator extracts geometric features between two amino acids by performing learnable vector calculations on their virtual atom coordinates. Specifically, when modeling the geometric relationship between two amino acids, such as the $i$-th and $j$-th amino acids, the input to the vector field operator consists of virtual atom groups, $\mathbf{Q}_i$ and $\mathbf{Q}_j$. To ensure that all vectors are represented in the same local coordinate system, we first transform $\mathbf{Q}_j$ from the local frame $\mathbf{T}_j$ to $\mathbf{T}_i$, and denote the transformed coordinates as $\mathbf{K}_j = \{\vec{\mathbf{k}}_k \in \mathbb{R}^3\}_{k=1,...,d_q}$, which can be written as (as shown in Figure 2.A ):

$$\mathbf{K}_j = \mathbf{T}_{i \leftarrow j} \circ \mathbf{Q}_j, \quad \text{where } \mathbf{T}_{i \leftarrow j} = \mathbf{T}_i^{-1} \circ \mathbf{T}_j \tag{1}$$

Here, $\mathbf{T}_{i \leftarrow j}$ represents the transformation matrix transforming the coordinates from $\mathbf{T}_j$ to $\mathbf{T}_i$. Next, to select specific virtual atoms for calculating feature vectors, similar to a linear layer, we introduce two sets of learnable weights, $\mathbf{w}^a = \{w_{k,l}^a \in \mathbb{R}\}_{k,l=1,...,d_q}$ and $\mathbf{w}^b = \{w_{k,l}^b \in \mathbb{R}\}_{k,l=1,...,d_q}$. Those weights are then utilized to perform vector calculations between $\mathbf{Q}_i$ and $\mathbf{K}_j$, resulting in extracted feature vectors $\mathbf{H}_{i,j} = \{\vec{\mathbf{h}}_k \in \mathbb{R}^3\}_{k=1,...,d_q}$ (as shown in Figure 2.B ). This can be expressed as follows:

$$\vec{\mathbf{h}}_k = \sum_l w_{k,l}^a \vec{\mathbf{q}}_l + \sum_l w_{k,l}^b \vec{\mathbf{k}}_l \tag{2}$$

where $\vec{\mathbf{q}}_l \in \mathbf{Q}_i, \vec{\mathbf{k}}_l \in \mathbf{K}_j, \vec{\mathbf{h}}_k \in \mathbf{H}_{i,j}$. Here, $\mathbf{H}_{i,j}$ serves as a vector representation for captured geometric features. However, due to its large numerical range (ranging from $-200\text{Å}$ to $200\text{Å}$), it can lead to instability during network training and requires further processing.

To avoid this issue, $\mathbf{H}_{i,j}$ is decomposed into two variables: unit direction vectors and vector lengths. The vector lengths will be mapped using a radial basis function, RBF. This can be written as:

$$\mathbf{g}_{i,j} = \text{concat}_k \left( \frac{\vec{\mathbf{h}}_k}{\|\vec{\mathbf{h}}_k\|}, \text{RBF}(\|\vec{\mathbf{h}}_k\|) \right), \quad \mathbf{g}_{i,j} \in \mathbb{R}^{d_g} \tag{3}$$

$\mathbf{g}_{i,j}$ is a vector that represents the geometric relationship between two residues and is used in the following module for aggregating and updating the features $\mathbf{s}_i$ and $\mathbf{e}_{i,j}$. The $\text{concat}_k$ represents the concatenation of all the feature[1] resulting from all the vectors in $\mathbf{H}_{i,j} = \{\vec{\mathbf{h}}_k\}_{k=1,...,d_q}$. $d_g$ represents the number of channels in $\mathbf{g}_{i,j}$.

## 3.3 Node Interactions

Here, a MLP-based multi-head attention mechanism is designed to aggregate geometric features $\mathbf{g}_{i,j}$, node features $\mathbf{s}_i$ and $\mathbf{s}_j$, edge features $\mathbf{e}_{i,j}$, and update the node representations $\mathbf{s}_i$. Specifically, the pair-wise features mentioned above are first fed into an MLP, followed by a softmax operation to obtain attention weights, denoted as:

$$a_{i,j} = \text{softmax}_j(\text{MLP}(\mathbf{s}_i, \mathbf{s}_j, \mathbf{g}_{i,j}, \mathbf{e}_{i,j})), \tag{4}$$

---

[1]representing the flattened unit direction vectors and the values output by RBF

where $a_{i,j}$ represents the attention weight for the interaction between nodes $i$ and $j$. Next, another MLP is employed to generate the values $\mathbf{v}_{i,j}$ for the multi-head attention, which are subsequently aggregated using the attention mechanism, expressed as follows:

$$\mathbf{o}_i = \sum_j a_{i,j} \mathbf{v}_{i,j}, \quad \text{where } \mathbf{v}_{i,j} = \text{MLP}(\mathbf{s}_j, \mathbf{g}_{i,j}, \mathbf{e}_{i,j}) \tag{5}$$

Here, $\mathbf{o}_i$ represents the aggregated features, which are utilized to update the features $\mathbf{s}_i$ after undergoing an MLP layer, written as:

$$\mathbf{s}_i \leftarrow \mathbf{s}_i + \text{MLP}(\mathbf{o}_i) \tag{6}$$

Up to this point, node features $\mathbf{s}_i$ have been updated and are utilized as the input for the subsequent layer and following operations.

### 3.4 Edge Interactions

Next, we introduce the edge interactions, which is designed to aggregate geometric information $\mathbf{g}_{i,j}$, node features $\mathbf{s}_i$ and $\mathbf{s}_j$, and edge features $\mathbf{e}_{i,j}$, to update the representation of the edge $\mathbf{e}_{i,j}$. This can be written as follows:

$$\mathbf{e}_{i,j} \leftarrow \mathbf{e}_{i,j} + \text{MLP}(\mathbf{s}_i, \mathbf{s}_j, \mathbf{g}_{i,j}, \mathbf{e}_{i,j}) \tag{7}$$

### 3.5 Virtual Atom Coordinates Updating

In the final stage of each VFN layer, the coordinates of virtual atoms $\mathbf{Q}_i$ are updated. We have devised two different methods for updating these virtual atom coordinates, which are respectively referred to as 'node feature-based updating' and 'coordinate aggregating updating.' These two approaches can be selected based on the specific task, and their detailed methodologies are elucidated following.

**Node Feature-Based Updating**. Node features $\mathbf{s}_i$ are processed through a linear layer to generate a set of virtual atom coordinates for updating $\mathbf{Q}_i$. This process can be represented as follows:

$$\mathbf{Q}_i \leftarrow \text{Linear}(\mathbf{s}_i) \tag{8}$$

**Coordinate Aggregating Updating**. For updating $\mathbf{Q}_i$, virtual atom coordinates $\mathbf{K}_j$ are firstly aggregated through an attention mechanism to obtain the aggregated atom coordinates $\mathbf{Q}_i^{\text{o}}$. Subsequently, $\mathbf{Q}_i^o$ is fed into an MLP layer to update the coordinates $\mathbf{Q}_i$, denoted as follows:

$$\mathbf{Q}_i \leftarrow \text{V-MLP}(\mathbf{Q}_i, \mathbf{Q}_i^{\text{o}}), \quad \text{where } \mathbf{Q}_i^{\text{o}} = \sum_j a_{i,j} \mathbf{K}_j \tag{9}$$

Here, V-MLP represents a dedicated MLP layer designed specifically for vectors, as described in Appendix A.2.1; $a_{i,j}$ is computed in Eq. equation 4.

## 4 Implementation for De novo Protein Design

To establish the recent paradigm in *de novo* protein design, we have developed two distinct models, namely VFN-Diff and VFN-IF, each dedicated to protein structure diffusion and inverse folding, respectively. In the protein diffusion part, the protein structure is designed and represented using backbone frames $\mathcal{T}$. Subsequently, these backbone frames are fed into the inverse folding network to obtain the corresponding protein sequence for the designed structure. In the following subsections, we present an overview of the implementations for VFN-Diff and VFN-IF. For more implementation details, please refer to Appendix A.2.2 and Appendix A.2.3.

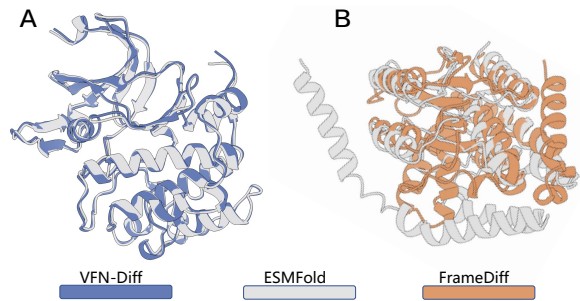

Figure 3: Visual Comparison of VFN-IF and Frame-Diff. 'ESMFold' represents protein structures recovered using ProteinMPNN and ESMFold, with closer structural resemblance being preferable.

### 4.1 PROTEIN DIFFUSION

We adopted the FrameDiff (Yim et al., 2023) paradigm, a diffusion model used to sample protein backbone structures by updating residue frames $\mathcal{T}$. In FrameDiff, a network called $\mathrm{FramePred}$ is employed to model the protein backbone frames $\mathcal{T}$ during each diffusion step. This network relies on invariant point attention (IPA), which consists of three components: node attention, edge attention, and point attention. Importantly, the point attention module is the operation that causes the *atom representation bottleneck*, as mentioned in §2.4. Therefore, to fairly evaluate whether VFN can bypass this bottleneck, we replaced the point attention with our VFN attention mechanism. The remaining parts of VFN-Diff remain consistent with FrameDiff.

### 4.2 INVERSE FOLDING

The purpose of the inverse folding task is to map the frames $\mathcal{T}$ (generated by the diffusion model) to amino acid categories $\boldsymbol{c} \in \{1, ..., 20\}^n$, denoted as $f_{\mathrm{if}} : \mathcal{T} \to \boldsymbol{c}$. These predicted amino acid categories aim to enable the protein to fold into the designed structure. In this task, we adopt the previous paradigm. Specifically, the VFN-IF network is composed of 15 sequential VFN layers, and it constructs edges in the graph using a $k$-NN approach. The network takes backbone frames and backbone atoms as input. It's important to note that the coordinates of backbone atoms are used to initialize some of the virtual atom coordinates, enabling VFN to achieve higher modeling capacity through hierarchical representation. The prediction head of VFN-

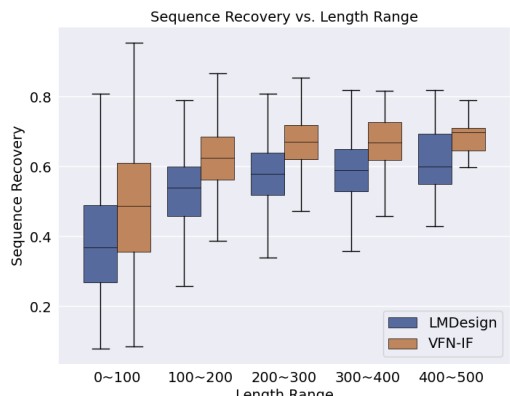

Figure 4: Visualization comparison, conducted on inverse folding and CATH 4.2, between sequence recovery rate and protein length.

IF follows the PiFold approach (Gao et al., 2022), directly predicting amino acid categories for each node through a linear layer and supervising them using a cross-entropy loss function. Additionally, we propose an approach to fine-tune the ESM model (Lin et al., 2023) with LoRA (Hu et al., 2021) to correct the predictions of VFN-IF. We name this approach VFN-IFE, and the process can be represented as $f_{\mathrm{ESM}} \circ f_{\mathrm{if}}(\mathcal{T}) : \mathcal{T} \to \boldsymbol{c}$, where $f_{\mathrm{if}}$ represents VFN-IF, and $f_{\mathrm{ESM}} : \boldsymbol{c}^{\mathrm{o}} \to \boldsymbol{c}$ denotes the ESM model fine-tuned with LoRA to correct the VFN-IF predictions $\boldsymbol{c}^{\mathrm{o}}$. For more VFN-IFE details, please refer to Appendix A.2.4.

## 5 EXPERIMENTS

As mentioned, we comprehensively validated the superiority of VFN-IF and VFN-Diff through experiments involving protein diffusion and inverse folding. The following sections will present the main experimental results for these two tasks separately. Additionally, we have included experiment details, extensive visual analyses and ablation studies based on inverse folding in Appendix A.4. We encourage readers to refer to the appendix for more in-depth information.

### 5.1 INVERSE FOLDING

In inverse folding, we followed the settings of (Gao et al., 2022) and tested the sequence recovery performance of VFN on the CATH 4.2 (Orengo et al., 1997), TS50, and TS500 datasets (Li et al., 2014). Furthermore, we tested the structure recovery performance of VFN and compared it with state-of-the-art models. Additionally, we demonstrated the superiority of VFN in trades-off between speed and accuracy.

In this task, we have devised three variations of the VFN model, namely VFN-IF, VFN-IF+, and VFN-IFE. VFN-IF represents the vanilla version of VFN, trained solely on the CATH 4.2 dataset. VFN-IF+ is an extended version of VFN-IF, scaled up to incorporate the entire PDB dataset during training. VFN-IFE denotes the version of VFN-IF equipped with an external knowledge base (ESM).

| | Model | Perplexity↓ | | | Recovery(%) ↑ | | |
|---|---|---|---|---|---|---|---|
| | | Short | Single | All | Short | Single | All |
| w/o ESM | StructGNN | 8.29 | 8.74 | 6.40 | 29.44 | 28.26 | 35.91 |
| | GraphTrans | 8.39 | 8.83 | 6.63 | 28.14 | 28.46 | 35.82 |
| | GCA | 7.09 | 7.49 | 6.05 | 32.62 | 31.10 | 37.64 |
| | GVP | 7.23 | 7.84 | 5.36 | 30.60 | 28.95 | 39.47 |
| | GVP-large† | 7.68 | 6.12 | 6.17 | 32.60 | 39.40 | 39.20 |
| | AlphaDesign | 7.32 | 7.63 | 6.30 | 34.16 | 32.66 | 41.31 |
| | ESM-IF† | 8.18 | 6.33 | 6.44 | 31.30 | 38.50 | 38.30 |
| | ProteinMPNN | 6.21 | 6.68 | 4.61 | 36.35 | 34.43 | 45.96 |
| | PiFold | 6.04 | 6.31 | 4.55 | 39.84 | 38.53 | 51.66 |
| | VFN-IF | **5.70** | **5.86** | **4.17** | **41.34** | **40.98** | **54.74** |
| ESM | ESM-IF† | 6.05 | **4.00** | 4.01 | 38.10 | 51.50 | 51.60 |
| | LM-Design | 6.77 | 6.46 | 4.52 | 37.88 | 42.47 | 55.65 |
| | VFN-IFE | **4.92** | 4.22 | **3.36** | **50.00** | **52.13** | **62.67** |

Table 1: Experimental results comparison on the CATH dataset (inverse folding). Some results are reproduced by Gao et al. (2022). "†" denotes that the version of CATH used is 4.3, while for the remaining methods, the CATH version is 4.2.

**Sequence recovery.** The performance of VFN on the CATH dataset and TS50, TS500 is presented in Table 1, Table 3 and Figure 4, respectively. In these experiments, VFN is compared to other advanced models, such as StructGNN (Ingraham et al., 2019), GraphTrans (Ingraham et al., 2019), GCA (Tan et al., 2022), GVP (Jing et al., 2020), ESM-IF (Hsu et al., 2022), ProteinMPNN (Dauparas et al., 2022), PiFold (Gao et al., 2022), LM-design (Zheng et al., 2023), in terms of perplexity and sequence recovery performance.

| Metric | w/o ESM | | ESM | |
|---|---|---|---|---|
| | PiFold | VFN-IF | LM-Design | VFN-IFE |
| scTM > 0.5 | 90.98% | **92.37%** | 89.42% | **93.29%** |
| scRMSD < 2 | 60.35% | **62.89%** | 58.41% | **64.16%** |

Table 2: Experimental results on structure recovery (inverse folding). 'scTM > 0.5' represents the percentage of designed proteins that exhibit a structural similarity exceeding 0.5 with the desired protein. The same applies to 'scRMSD < 2'.

Table 1 provides detailed results for various subsets. Among the subsets considered in our analysis, the "Short" subset refers to proteins with a length of up to 100 amino acids, while the "Single" subset exclusively includes single chain proteins. 'w/o ESM' and 'ESM' refer to methods without and with the use of an external knowledge base (ESM), respectively.

**Structure recovery.** We compared VFN's performance in terms of protein structure recovery with several advanced methods on CATH 4.2, as shown in Table 2. We followed standard evaluation procedures(Yim et al., 2023). Specifically, we used ESMFold to predict whether sequences designed by inverse folding networks could fold proteins into the desired structures, i.e., the input structures of the inverse folding network. We employed two metrics, scTM↑ and scRMSD↓, to assess the similarity between the desired protein structures and the structures of proteins de-

| | Model | TS50 | | TS500 | |
|---|---|---|---|---|---|
| | | Perp.↓ | Rec.(%)↑ | Perp.↓ | Rec.(%)↑ |
| w/o ESM | StructGNN | 5.40 | 43.89 | 4.98 | 45.69 |
| | GraphTrans | 5.60 | 42.20 | 5.16 | 44.66 |
| | GVP | 4.71 | 44.14 | 4.20 | 49.14 |
| | GCA | 5.09 | 47.02 | 4.72 | 47.74 |
| | AlphaDesign | 5.25 | 48.36 | 4.93 | 49.23 |
| | ProteinMPNN | 3.93 | 54.43 | 3.53 | 58.08 |
| | PiFold | 3.86 | 58.72 | 3.44 | 60.42 |
| | VFN-IF | **3.58** | **59.54** | **3.19** | **63.65** |
| ESM | LM-Design | 3.50 | 57.89 | 3.19 | 67.78 |
| | VFN-IFE | **2.52** | **73.30** | **2.54** | **72.49** |

Table 3: Experimental results on TS50 and TS500 (inverse folding).

signed through inverse folding. Our experimental results demonstrate that VFN has a significant advantage compared to state-of-the-art methods.

**Speed and accuracy trade-off.** In terms of the trade-off between speed and accuracy, we compared VFN-IF with the SoTA model in this regard, PiFold. We conducted comparisons between different layers of VFN and PiFold, as shown in Figure 1(A). Experimental results demonstrate that VFN-IF achieves SoTA efficiency. Even with just 5 layers, VFN-IF achieves higher accuracy (52.68%) than a 10-layer PiFold while maintaining faster inference speeds. Furthermore, PiFold's accuracy saturates at 10 layers, whereas VFN-IF does not encounter this issue. For more details, please refer to Table 6a in Appendix.

## 5.2 PROTEIN DIFFUSION

We followed the settings and benchmarks of FrameDiff (Yim et al., 2023), conducting a detailed comparison between VFN-Diff and FrameDiff in terms of designability and diversity. It is worth emphasizing that FrameDiff and VFN-Diff differ only in the protein structure encoder (VFN vs. IPA). All other settings are identical, and the parameter counts are similar (18.3M vs. 17.4M), making it an ablation study. On the other hand, RFDiffusion (Watson et al., 2023) is a recent advance in the field. However, as FrameDiff pointed out, RFDiffusion performs noticeably worse than FrameDiff in the same setting (without pre-trained weights). Additionally, RFDiffusion has a larger number of parameters and is trained on larger datasets (i.e., complex data). Therefore, comparing with RFDiffusion is beyond the scope of this work. We leave these engineering implementations for future research.

**Designability.** As shown in Table 4 and Figure 5, we compare the protein designability of VFN-Diff with that of FrameDiff. For evaluation, Proteins generated by the diffusion model are reconstructed using the inverse folding network (ProteinMPNN) and structural prediction network (ESMFold). The designability of the generated proteins is then assessed by comparing their structural similarity (scTM, scRMSD) to the reconstructed proteins. Experimental results demonstrate that VFN-Diff outperforms FrameDiff noticeably in terms of designability.

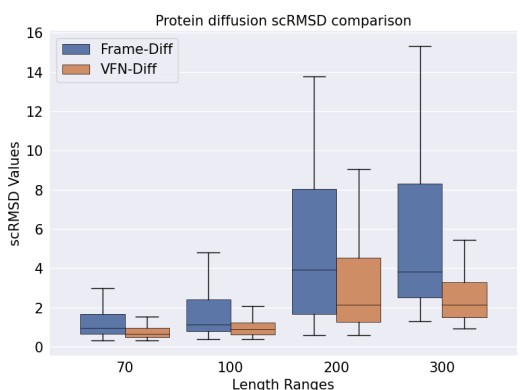

**Diversity.** Diversity is an important metric for generative models. Similarly, we follow FrameDiff's relevant evaluation metrics, namely 'diversity' and 'pdbTM', to compare the diversity of VFN-IF and FrameDiff. Specifically, 'diversity' represents the clustering center density of generated samples.

Figure 5: Visualization of designability. $N_{\text{seq}}$ represents the number of attempts when reconstructing protein structures using Protein MPNN and ESM-Fold.

To be more specific, we first excluded undesignable proteins (scTM < 0.5). Then, we used MaxCluster (Herbert & Sternberg, 2008) to cluster the remaining samples and obtain clustering centers. Finally, clustering center density can be calculated as follows: (number of clustering center) / (number of generated samples). 'pdbTM' represents the structural similarity of generated samples to the most similar structures in the PDB database.

**Visualization Comparison.** Through visualization, we observed that FrameDiff suffers from very low designability when designing longer protein, while VFN-Diff does not have this issue. To illustrate, we selected typical generated proteins and compared the results generated by VFN-Diff and FrameDiff through visualization, respectively. The visualization results demonstrate the superiority of VFN-Diff in generating protein, as shown in Figure 3.

|  |  | Setting | Noise Scale | 1.0 | 0.5 | 0.1 | 0.1 | 0.1 |
|---|---|---|---|---|---|---|---|---|
|  |  |  | Num. Step | 500 | 500 | 500 | 500 | 100 |
|  |  | Metric | Num. Seq. | 8 | 8 | 8 | 100 | 8 |
| Designability | scTM$_{0.5}$ ↑ |  | FrameDiff | 53.58% | 76.42% | 77.41% | 87.04% | 76.67% |
|  |  |  | VFN-Diff | **67.04%** | **81.23%** | **83.95%** | **92.84%** | **83.83%** |
|  | scRMSD$_2$ ↑ |  | FrameDiff | 10.62% | 23.46% | 28.02% | 37.78% | 26.42% |
|  |  |  | VFN-Diff | **25.93%** | **40.00%** | **44.20%** | **56.30%** | **40.25%** |
| Diversity | Diversity ↑ |  | FrameDiff | 51.98% | 74.57% | 75.56% | 85.43% | 74.94% |
|  |  |  | VFN-Diff | **66.54%** | **80.49%** | **83.33%** | **90.61%** | **82.59%** |
|  | pdbTM$_{0.7}$ ↑ |  | FrameDiff | 5 | 30 | 37 | 86 | 35 |
|  |  |  | VFN-Diff | **9** | **47** | **54** | **102** | **48** |

Table 4: Experimental results on protein structure diffusion assessing the designability and diversity of VFN-Diff. 'scTM$_{0.5}$' and 'scRMSD$_2$' represent the percentages of generated proteins with scTM > 0.5 and scRMSD < 2, respectively. 'pdbTM$_{0.7}$' signifies the count of generated proteins with pdbTM < 0.7, measuring the novelty of the generated protein. For more details on metrics, please refer to appendix A.4.3.

## 6 CONCLUSION

By introducing VFN, we address the atom representation bottleneck though the vector field operator, enhancing its capacity for modeling frames. We demonstrate VFN's expressiveness through comprehensive experiments in *de novo* protein design. Significant improvements or state-of-the-art performance are achieved in protein diffusion and inverse folding tasks.

**Ethics Statement** We do not foresee any obvious undesirable ethical or social impacts now.

## ACKNOWLEDGEMENT

This work was supported by National Key R&D Program of China (No. 2022ZD0118700). The authors would like to thanks Hangzhou City University for accessing its GPU cluster.

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

# A APPENDIX

## A.1 THE COMPARISON BETWEEN VFN AND IPA

In this section, we first present the pseudocode for IPA and VFN to provide an intuitive comparison, as shown in Subsection A.1.1. Subsequently, we elaborate on the differences between IPA and VFN, as outlined in Subsection A.1.2. In order to further elucidate the atom representation bottleneck, we expound on how VFN addresses the bottleneck in Subsection A.1.3. Finally, in Subsection A.1.4, we introduce the pipeline of IPA to facilitate our readers' understanding of the specific processes involved in IPA.

### A.1.1 THE PSEUDO-CODE FOR VFN AND IPA

In this section, we initially compare IPA with VFN in the provided Algorithm 1 and 2 below and explain the meaning of the notation. Please note that the following code is provided for the convenience of our readers to distinguish between IPA and VFN's core differences. The specific implementation of VFN may vary across different tasks and may differ from Algorithm 2. Please refer to the main text for the exact implementation.

---

**Algorithm 1** The pseudo-code for the IPA module.

1: def $\text{IPA}(\{\mathbf{s}_i\}, \{\mathbf{e}_{ij}\}, \{\mathbf{T}_i\})$:
2:      $\mathbf{q}_i, \mathbf{k}_i, \mathbf{v}_i = \text{Linear}(\mathbf{s}_i)$
3:      $\vec{\mathbf{q}}_{il}, \vec{\mathbf{k}}_{il}, \vec{\mathbf{v}}_{il} = \text{Linear}(\mathbf{s}_i)$
4:      $\vec{\mathbf{q}}_{il}, \vec{\mathbf{k}}_{il}, \vec{\mathbf{v}}_{il} \leftarrow \mathbf{T}_i \circ \{\vec{\mathbf{q}}_{il}, \vec{\mathbf{k}}_{il}, \vec{\mathbf{v}}_{il}\}$
5:      $a_{ij}^{\text{point}} = \sum_l \|\vec{\mathbf{q}}_{il} - \vec{\mathbf{k}}_{jl}\|$
     –
6:      $a_{ij}^{\text{node}} = \mathbf{q}_i^\top \mathbf{k}_j$
7:      $a_{ij}^{\text{edge}} = \text{Linear}(\mathbf{e}_{ij})$
8:      $a_{i,j} = \text{softmax}_j(a_{ij}^{\text{node}} + a_{ij}^{\text{edge}} + a_{ij}^{\text{point}})$
     –
9:      $\mathbf{o}_{ij}^{\text{node}} = \sum_j a_{i,j} \mathbf{v}_i$
10:      $\mathbf{o}_{ij}^{\text{edge}} = \sum_j a_{i,j} \mathbf{e}_{ij}$
11:      $\vec{\mathbf{o}}_{il}^{\text{point}} = \mathbf{T}_i^{-1} \circ \sum_j a_{i,j} \vec{\mathbf{v}}_{jl}$
12:      $\mathbf{s}_i \leftarrow \text{Linear}(\mathbf{o}_{ij}^{\text{edge}}, \mathbf{o}_{ij}^{\text{node}}, \vec{\mathbf{o}}_{il}^{\text{point}})$
     –
13:      return $\{\mathbf{s}_i\}$

---

**Algorithm 2** The pseudo-code for the VFN module.

1: def $\text{VFN}(\{\mathbf{s}_i\}, \{\mathbf{e}_{ij}\}, \{\mathbf{T}_{i \leftarrow j}\})$:
     –
2:      $\vec{\mathbf{q}}_{il} = \text{Linear}(\mathbf{s}_i)$
3:      $\vec{\mathbf{k}}_{jl} = \mathbf{T}_{i \leftarrow j} \circ \vec{\mathbf{q}}_{jl}$
4:      $\vec{\mathbf{h}}_k = \sum_l w_{kl}^{\text{a}} \vec{\mathbf{q}}_{il} + \sum_l w_{kl}^{\text{b}} \vec{\mathbf{k}}_{jl}$
5:      $\mathbf{g}_{i,j} = \text{concat}_k \left( \frac{\vec{\mathbf{h}}_k}{\|\vec{\mathbf{h}}_k\|}, \text{RBF}(\|\vec{\mathbf{h}}_k\|) \right)$
     –
     –
6:      $a_{i,j} = \text{softmax}_j(\text{MLP}(\mathbf{s}_i, \mathbf{s}_j, \mathbf{g}_{i,j}, \mathbf{e}_{i,j}))$
7:      $\mathbf{v}_{i,j} = \text{MLP}(\mathbf{s}_j, \mathbf{g}_{i,j}, \mathbf{e}_{i,j})$
8:      $\mathbf{o}_i = \sum_j a_{i,j} \mathbf{v}_{i,j}$
     –
     –
9:      $\mathbf{s}_i \leftarrow \mathbf{s}_i + \text{MLP}(\mathbf{o}_i)$
10:      $\mathbf{e}_{i,j} \leftarrow \text{MLP}(\mathbf{s}_i, \mathbf{s}_j, \mathbf{g}_{i,j}, \mathbf{e}_{i,j})$
11:      return $\{\mathbf{s}_i\}, \{\mathbf{e}_{i,j}\}$

---

In the aforementioned pseudocode, we largely adhere to the notations used in the main text, with some distinctions in certain notations. Specifically, $\vec{\mathbf{q}}_{il}$ denotes the $l$-th feature vector of the $i$-th node, and $\vec{\mathbf{k}}_{jl}$ follows the same convention. $\mathbf{T}_i$ represents the transformation matrix from the $i$-th local frame (residue frame) to the global frame, while $\mathbf{T}_i^{-1}$ signifies the transformation matrix from the global frame to the $i$-th local frame. $\mathbf{T}_{i \leftarrow j}$ denotes the transformation matrix from the $j$-th local frame to the $i$-th local frame. $\mathbf{s}_i$ represents the representation of the $i$-th node, and $\mathbf{e}_{ij}$ represents the representation of the edge between $i$-th and $j$-th nodes. $w_{kl}^{\text{a}}$ and $w_{kl}^{\text{b}}$ are learnable weights. The specific process of VFN is detailed in the main text; the procedure for IPA is elaborated in Subsection A.1.4. For the sake of conciseness in comparison, the pseudocode overlooks certain non-essential factors. For instance, in the pseudocode, we illustrate a single-head attention mechanism, but in fact, both IPA and VFN employ a multi-head attention mechanism.

### A.1.2 THE DIFFERENCES BETWEEN IPA AND VFN

As indicated in the pseudocode, the attention mechanism for virtual atoms in IPA and VFN is fundamentally different. The most crucial distinction lies in the fact that, due to the constraints of SE(3) invariance, IPA cannot directly employ an activation function when extracting features for virtual atoms, as explained in Subsection A.1.3. In contrast, VFN circumvents this limitation and utilizes a vector field operator to extract features, denoted as $\vec{\mathbf{h}}_k$, and complementing it with an MLP (ReLU inside) for feature extraction. To accommodate this design choice, the overall architecture of VFN diverges significantly from that of IPA.

### A.1.3 BYPASSING IPA BOTTLENECKS

The design of IPA has led to the atom representation bottleneck, while the design of VFN avoids this issue. Specifically, in IPA, $\mathbf{q}_i$ and $\mathbf{k}_j$ are with respect to the global frame, and the operator corresponding to $\mathbf{q}_i, \mathbf{k}_j$ needs to maintain SE(3) invariance. This constraint results in the inability to directly apply activation functions on $\mathbf{q}_i, \mathbf{k}_j$, as doing so would compromise the SE(3) invariance. Consequently, IPA can only employ operations similar to distance pooling. We refer to this limitation as the atom representation bottleneck.

In VFN, we place these virtual atoms in the same local frame $\mathbf{T}_i$, utilizing the local frame to ensure SE(3) invariance, as proved in Subsection A.2.6. This eliminates the need to impose constraints on operators to achieve SE(3) invariance. Operators in VFN can freely utilize activation functions without disrupting SE(3) invariance. This characteristic allows VFN to circumvent the atom representation bottleneck present in IPA.

### A.1.4 THE PIPELINE OF IPA

As shown in Algorithm 1, IPA employs an attention mechanism. Unlike VFN, its attention mechanism consists of three parts: node attention, edge attention, and point attention. Specifically, node attention and edge attention utilize common methods. In particular, node attention employs a mechanism similar to that of the transformer, obtaining attention weights $a_{ij}^{\text{node}}$ through dot product of $\mathbf{q}_i$ and $\mathbf{k}_j$. In edge attention, the representation of edges $\mathbf{e}_{ij}$ introduces attention bias $a_{ij}^{\text{edge}}$ through a linear layer. Point attention is the core of IPA, where attention weights $\vec{\mathbf{o}}_{il}^{\text{point}}$ are obtained through a distance pooling operation on the virtual atomic distances ($\|\vec{\mathbf{q}}_{il} - \vec{\mathbf{k}}_{jl}\|$). Subsequently, these attention weights are summed and normalized through softmax to obtain the final attention weights $a_{i,j}$. These attention weights $a_{i,j}$ are then used to calculate weighted averages for the corresponding representations $\mathbf{v}_i, \mathbf{e}_{ij}, \vec{\mathbf{v}}_{jl}$, producing the output for each type of attention. $\mathbf{o}_{ij}^{\text{node}}, \mathbf{o}_{ij}^{\text{edge}}, \vec{\mathbf{o}}_{il}^{\text{point}}$. Finally, these outputs are collectively updated for each node's representation $\mathbf{s}_i$ through a linear layer, achieving the frame modeling.

## A.2 METHODOLOGY

### A.2.1 MULTI-LAYER VECTOR PERCEPTRON MODULE

In this section, we elaborate on the process of the multi-layer vector perceptron module, V-MLP, as illustrated in Algorithm 3:

---

**Algorithm 3** The pseudo-code for the multi-layer vector perceptron module.

1: def V-MLP($\mathbf{Q}_i, \mathbf{Q}_i^o$):

2:      $\mathbf{w}^c = \{w_{k,l}^c \in \mathbb{R}\}_{k,l=1,...,d_q}, \mathbf{w}^d = \{w_{k,l}^d \in \mathbb{R}\}_{k,l=1,...,d_q}$

     *# initialize learnable weight* $\mathbf{w}^c, \mathbf{w}^d$

3:      $\vec{\mathbf{v}}_k = \sum_l w_{k,l}^c \vec{\mathbf{q}}_l + \sum_l w_{k,l}^d \vec{\mathbf{q}}_l^o$                  $\vec{\mathbf{v}}_k \in \mathbb{R}^3 , \vec{\mathbf{q}}_l \in \mathbf{Q}_i, \vec{\mathbf{q}}_l^o \in \mathbf{Q}_i^o$

4:      $\vec{\boldsymbol{u}}_k = \frac{\vec{\mathbf{w}}_k \cdot \vec{\mathbf{v}}_k}{\|\vec{\mathbf{w}}_k\| \|\vec{\mathbf{v}}_k\|} \vec{\mathbf{v}}_k$                  $\vec{\mathbf{w}}_k \in \mathbb{R}^3, \ (\vec{\mathbf{w}}_k \cdot \vec{\mathbf{v}}_k) \in \mathbb{R}$

     *#* $\vec{\mathbf{w}}_k$ *are learnable weights.*

5:      $\mathbf{w}^e = \{w_{m,k}^e \in \mathbb{R}\}_{m,k=1,...,d_q}$

     *# initialize learnable weight* $\mathbf{w}^e$.

6:      $\vec{\boldsymbol{e}}_m = \sum_k w_{m,k}^e \vec{\boldsymbol{u}}_k$                           $\vec{\boldsymbol{e}}_m \in \mathbb{R}^3$

7:      return $\{\vec{\boldsymbol{e}}_m\}_{m=1,....,d_q}$

---

### A.2.2   IMPLEMENTATION OF VFN-DIFF

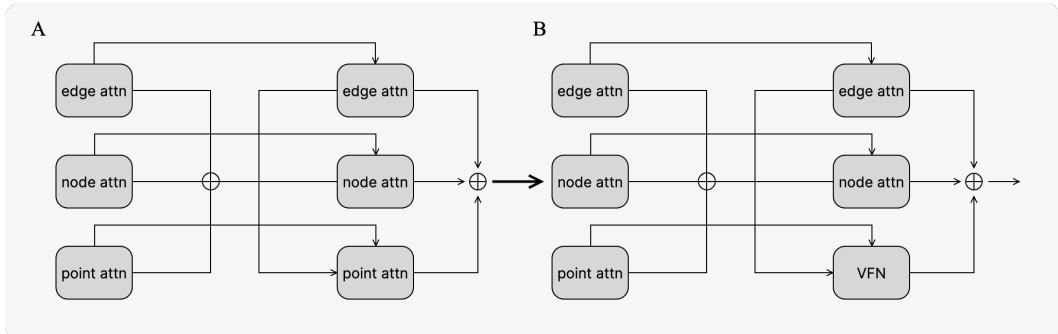

Figure 6: In the context of FrameDiff (A) and VFN-Diff (B), the traditional point attention module in FrameDiff has been replaced with a VFN layer, introducing enhanced geometric feature extraction capabilities

We have implemented VFN-Diff based on FrameDiff, as shown in Figure 6. Apart from modifications to the structural encoder, IPA, all other parts remain unchanged. Within IPA, attention is divided into three components: node attention, edge attention, and point attention. In the case of VFN-Diff, we replace the point attention component in IPA with the VFN layer, while keeping the remaining portions unaltered. Despite the removal of node attention and edge attention in VFN-Diff, the model still functions effectively, achieving performance comparable to FrameDiff. However, it is important to note that, at this stage, VFN-Diff's parameter count is only one-ninth that of FrameDiff. Clearly, such a comparison would be unfair. Therefore, we retained the additional parameter-rich "vanilla" components, node attention and edge attention. This adjustment brings the parameter count of VFN-Diff close to that of FrameDiff (18.3 million vs. 17.4 million), facilitating a fair comparison. Moreover, node attention and edge attention are well-established practices, widely adopted in previous works such as PiFold. Consequently, these modules do not constitute the core of IPA; rather, point attention does. Hence, the replacement of point attention with the VFN-Diff approach represents a rigorously comparative implementation.

### A.2.3   IMPLEMENTATION OF VFN-IF

We have adopted the PiFold framework to implement VFN, consisting of three components: the input layer, network layer, and decoder. Concerning the decoder and corresponding loss functions, VFN remains consistent with PiFold. For the network layer, we have substituted PiFold's layer, PiGNN,

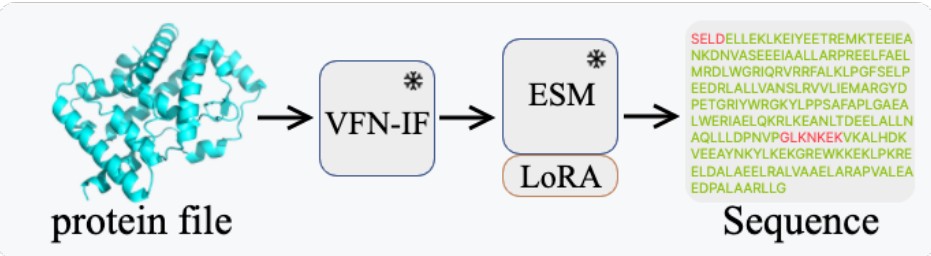

Figure 8: Flowchart depicting the fine-tuning process of frozen VFN-IF and ESM with LoRA.

with the VFN layer. However, to ensure a fair comparison, the Global Context Attention module from PiFold has also been retained in VFN-IF.

Regarding the network input layer, PiFold employs a manual featurizer to extract interatomic features such as angles and distances. In VFN, we have completely removed this featurizer since the VFN layer itself can extract geometric features without relying on a featurizer. We initialize a portion of virtual atomic coordinates using known backbone atomic coordinates to provide the network with this information.

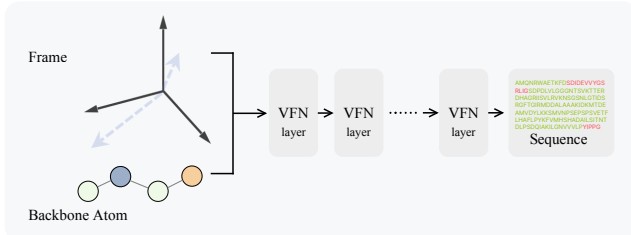

Figure 7: We harnessed the capabilities of VFN layers to effectively extract both frame and backbone atom information, facilitating the generation of sequences for inverse folding.

### A.2.4 IMPLEMENTATION OF VFN-IFE

In VFN-IFE, the ESM model is employed to refine the predictions made by VFN-IF. During training, the pretrained VFN-IF is frozen, and its predictions are used as input to the ESM model (as described later). The ESM model is initialized with pretrained weights and then frozen. We leverage LoRA to provide learnable weights and fine-tune the ESM. ESM is supervised with a cross-entropy loss using ground truth amino acid labels. The most critical aspect of our approach is the introduction of a relaxation method, allowing the ESM to accept probability-based inputs from VFN-IF.

For $i$-th amino acid, VFN-IF predicts a set of probabilities corresponding to 20 amino acid categories, denoted as $\boldsymbol{p} = \{0 < p_j < 1\}_{j=0,\dots,20}$. Similarly, ESM includes word embeddings corresponding to these categories, denoted as $\mathbf{W}_{\text{ESM}} = \{\mathbf{w}_j \in \mathbb{R}^{d_{\text{ESM}}}\}_{j=0,\dots,20}$. Therefore, we use the predicted probabilities $\boldsymbol{p}$ from VFN to perform a weighted sum of the corresponding word embeddings $\mathbf{W}_{\text{ESM}}$ for producing the input token $s_i^{\text{ESM}}$ to ESM, denoted as:

$$s_i^{\text{ESM}} = \sum_j p_j \mathbf{w}_j \tag{10}$$

### A.2.5 LOCAL FRAMES OF RESIDUES

In VFN, a local frame is set up for each residue via a Gram–Schmidt process proposed by AlphaFold2 (Jumper et al., 2021), refer to $\mathrm{rigidFrom3Points}$ algorithm in their paper.

### A.2.6 PROOF OF SE(3) INVARIANCE

The output and the vector representation of VFN are SE(3) invariant, which are crucial for networks to attain higher performance. The proof is simple. In short, the local frame of residues is SE(3) equivariant, which ensures the invariance for the inputs of VFN and the outputs of vector field operator. Here, we provide the proof following:

**The SE(3) invariance of initial representation.** Due to all the initial representations are with respect to the local frame, the input of VFN, $\mathbf{s}_i, \mathbf{e}_{i,j}, \vec{\mathbf{q}}_k$, is SE(3) invariant.

**The SE(3) invariance of the vector field operator.** In our main paper, the transform matrix $T_{i \leftarrow j}$ is employed in our operators (refer to Eq. equation 1 in our main paper):

$$\mathbf{K}_j = \mathbf{T}_{i \leftarrow j} \circ \mathbf{Q}_j, \quad \text{where } \mathbf{T}_{i \leftarrow j} = \mathbf{T}_i^{-1} \circ \mathbf{T}_j \tag{11}$$

The transform matrix $T_{i \leftarrow j}$ is SE(3) invariant *w.r.t.* the global reference frame, because the global frame cancels out in the computation of the transform matrix $\mathbf{T}_{i \leftarrow j}$:

$$
\begin{aligned}
(\mathbf{T}_{\text{global}} \circ \mathbf{T}_i)^{-1} \circ (\mathbf{T}_{\text{global}} \circ \mathbf{T}_j) &= \mathbf{T}_i^{-1} \circ \mathbf{T}_{\text{global}}^{-1} \circ \mathbf{T}_{\text{global}} \circ \mathbf{T}_j \\
&= \mathbf{T}_i^{-1} \circ \mathbf{T}_j
\end{aligned}
\tag{12}
$$

where $\mathbf{T}_{\text{global}}$ denotes any global reference frame. Therefore, the outputs $\vec{\mathbf{h}}_k, \mathbf{g}_{i,j}$ of the vector field operator are SE(3) invariant.

**The conclusion of SE(3) invariance.** Finally, because the input of VFN and the vector field operator (Only these operators associated with the global reference frame) are SE(3) invariant, the intermediate variables and outputs of VFN satisfy SE(3) invariance.

### A.3 DE NOVO PROTEIN DESIGN USING VFN

In this section, we explored *de novo* protein design through the utilization of two distinct pipelines. The first pipeline involves the established flow of FrameDiff coupled with Protein MPNN, while the alternative approach adpots VFN-Diff in tandem with VFN-IFE. Both pipleines employed the ESMFold (Hie et al., 2022) to fold amino acid sequences into ptrotein structures.Remarkably, in one-shot experiments with a Noise Scale parameter set to 1.0, our VFN-based pipeline outperformed FrameDiff+Protein MPNN approach in terms of designability, as shown in §A.4.3.

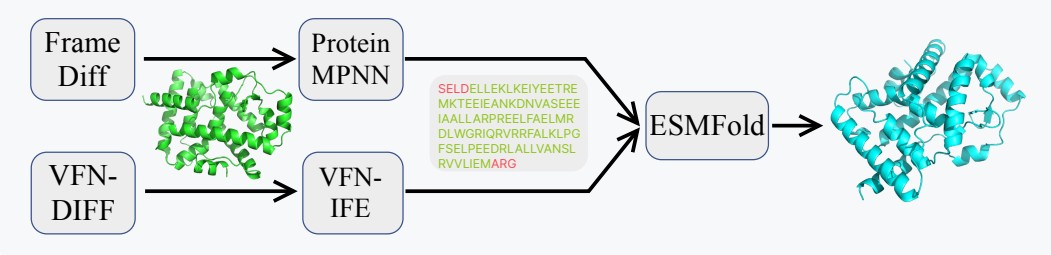

Figure 9: Whole pipelines

### A.4 EXPERIMENT

### A.4.1 DATASET DETAILS

**Protein diffsuion.** FrameDiff was re-trained using the same standards as in (Yim et al., 2023), while VFN-Diff was trained on four NVIDIA 4090 GPUs for a total duration of 10 days and 14 hours. The training dataset consisted of proteins from the PDB database (Berman et al., 2000) in August 2023, encompassing 21,399 proteins with lengths ranging from 60 to 512 and a resolution of $< 5\text{Å}$.

**Inverse folding.** Unless specified, our experiments are conducted on the CATH 4.2 (Orengo et al., 1997) dataset using the same data splitting as previous works such as GVP (Jing et al., 2020) and PiFold (Gao et al., 2022). The dataset consists of 18,024 proteins for training, 608 for validation, and

| | Setting | Noise Scale | 1.0 | 0.5 | 0.1 | 0.1 |
|---|---|---|---|---|---|---|
| | | Number Steps | 500 | 500 | 500 | 100 |
| | Metric | Number Sequences | 8 | 8 | 8 | 8 |
| **Designability** | $\text{scTM}_{0.5} \uparrow$ | FrameDiff + ProteinMPNN | 53.58% | 76.42% | 77.41% | 76.67% |
| | | VFN-Diff + ProteinMPNN | 67.04% | 81.23% | 83.95% | 83.83% |
| | | VFN-Diff + VFN-IF | **72.84%** | **91.60%** | **93.46%** | **90.49%** |
| | $\text{scRMSD}_2 \downarrow$ | FrameDiff + ProteinMPNN | 10.62% | 23.46% | 28.02% | 26.42% |
| | | VFN-Diff + ProteinMPNN | 25.93% | 40.00% | 44.20% | 40.25% |
| | | VFN-Diff + VFN-IF | **26.79%** | **53.33%** | **58.27%** | **51.36%** |
| **Diversity** | Diversity $\uparrow$ | FrameDiff + ProteinMPNN | 51.98% | 74.57% | 75.56% | 74.94% |
| | | VFN-Diff + ProteinMPNN | 66.54% | 80.49% | 83.33% | 82.59% |
| | | VFN-Diff + VFN-IF | **69.75%** | **86.91%** | **87.03%** | **85.43%** |

Table 5: Comparison of the complete pipeline. All settings are aligned with Table 4 in the main text. Here, VFN-IF adopts the settings of ProteinMPNN.

1120 for testing. During the evaluation, we also test our model on two smaller datasets, TS50 and TS500 (Jing et al., 2020; Gao et al., 2022), to validate the generalizability. Furthermore, we also create another larger training set by incorporating data from the PDB (Burley et al., 2017). We apply the same strategy as in (Zhou et al., 2023) to collect and filter structures. Additionally, the proteins with sequences highly similar to test set proteins are also removed. By using the expanded dataset, we are able to scale up the VFN-IF.

### A.4.2 IMPLEMENT DETAILS

**VFN-Diff.** Our training regimen employs the Adam optimizer with the following hyperparameters: a learning rate of 0.0001, $\beta_1$ set to 0.9, and $\beta_2$ set to 0.999. We follow the setting of FrameDiff. For more implement details, please refer to thier paper.

**VFN-IF.** VFN-IF are trained with batch size 8 and are optimized by AdamW with a weight decay of 0.1. We apply a OneCycle scheduler with a learning rate of 0.001 and train our model for a total of 100,000 iterations.

**VFN-IFE.** We constructed VFN-IFE by employing a 15B ESM model in conjunction with the standard VFN-iF. The rank for LoRA applied to ESM was set to 8. All other training settings remained consistent with those of VFN-iF.

### A.4.3 METRIC

**Protein diffusion inference.** Inferences were conducted with protein lengths ranging from 100 to 500, using a step size of 5. This resulted in the generation of 10 diffusion samples of protein at each step, with a total of 810 diffusion samples, denoted as $N_{diff}$. 'Noise Scale' represents the initial noise scale in diffusion, while 'Num. Step' represent the diffusion steps during inference. Protein MPNN was employed to generate 'Num. Seq.' ($N_{seq}$) sequences for each sample, followed by ESMFold to create protein structure files, referred to as $N_{esm} = N_{diff} \times N_{seq}$.

**Protein diffusion Metrics.** Metrics of protein diffusion inference experiments can be categorized into the following sections:

- **Structural Similarity Metrics:** This metric evaluates the proportion of samples with a structural consensus TM score (scTM) and RMSD meeting the criteria of scTM > 0.5 and scRMSD < 2 Å, as indicated in Table 4 by $\text{scTM}_{0.5}$ and $\text{scRMSD}_2$. In this context, scTM measures the structural similarity between ESMFold generated proteins and diffusion generated structures, while scRMSD quantifies the root-mean-square deviation in atomic positions between these structures.

- **Diversity:** To gauge the diversity of the generated protein sequences, we adhered to the methodology laid out in (Yim et al., 2023) and leveraged MaxCluster (Herbert & Sternberg, 2008) for hierarchical clustering of protein backbones. However, we opted for a higher threshold of 0.6 TM-score, in contrast to the 0.5 TM-score referenced in the literature, to

impose a more rigorous clustering criterion. Furthermore, we enforced a selection criterion for cluster inclusion, requiring scTM > 0.5, with the intention of mitigating the impact of proteins with low designability on diversity assessments. For each diffusion sample, a singular protein generated through ESMFold was chosen based on the highest scTM score. The diversity metric was computed as the ratio of the number of clusters to $N_{diff}$. These experimental modifications were introduced to yield results that are not only more scientifically sound but also align more closely with anticipated trends within the inference data.

- **pdbTM:** To assess protein novelty, we compared the ESMFold-generated PDB files, each containing the protein with the highest scTM score from a diffusion sample, to the PDB database using the Foldseek tool (van Kempen et al., 2024). The highest TM-scores of the generated samples were compared with any chain in the PDB database, and the resulting value was denoted as pdbTM. To exclude proteins with limited designability, we applied a cutoff criterion of RMSD < 2, consistent with the approach used in (Yim et al., 2023). pdbTM served as a robust metric for quantifying protein novelty, reflecting the structural similarity between the generated proteins and those documented in the PDB database. In contrast to the threshold of pdbTM < 0.6 employed in (Yim et al., 2023), we considered proteins with pdbTM values less than 0.7, which refers as $pdbTM_{0.7}$ in Table 4, as novel designs due to their substantial dissimilarity from existing proteins, resulting in the inclusion of a greater number of novel proteins in both selection.

**Inverse folding.** We conducted a comparative analysis involving VFN-IF, VFN-IFE, Pifold, and LM-Design. We utilized the CATH 4.2 dataset for validation and compared the results by performing ESMFold on the generated sequences (with $N_{seq} = 1$) and comparing them with the original PDB files to calculate scTM and scRMSD. This analysis was performed as a one-shot comparison.

**Whole Pipeline.** This set of experiments compared VFN-Diff + VFN-IFE with FrameDiff + Protein-MPNN. To ensure alignment of comparison standards, we used $N_{seq} = 1$ and Noise Scale = 1.0 for one-shot comparisons.

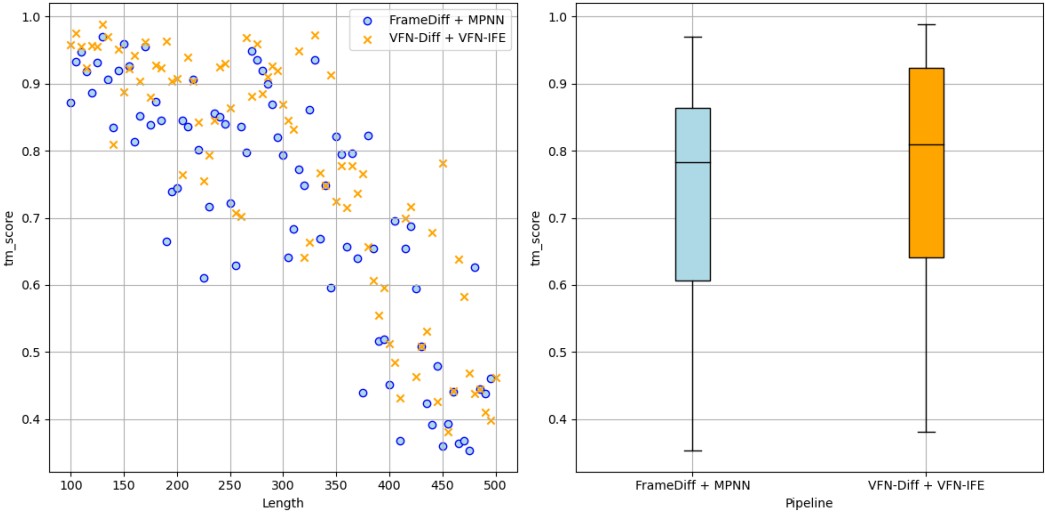

Figure 10: Whole pipeline in one-shot: FrameDiff+ProteinMPNN and VFN-Diff+VFN-IFE

### A.4.4 ABLATION STUDY

### A.5 ABLATIONS

In this section, we carefully investigate the design choices of vector field modules proposed here based on CATH 4.2 (inverse folding).

**The number of layers.** We investigate the impact of modifying the number of layers on recovery and perplexity in Table 6a. Increasing the number of layers from 5 to 15 results in a marginal improvement

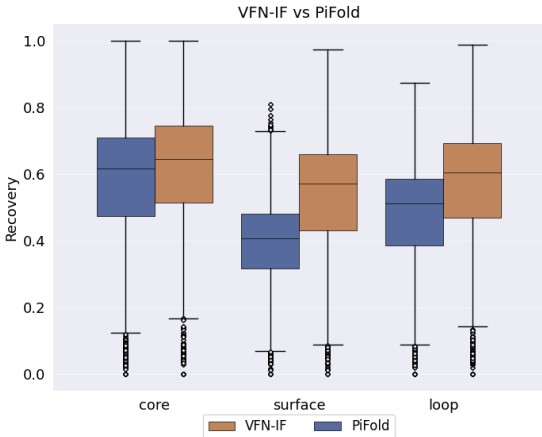

Figure 11: VFN vs PiFold on different structural contexts

Table 6: Ablation studies on the CATH 4.2 dataset. We use the default model settings unless otherwise specified. When calculating the number of parameters, we only count the number of parameters occupied by this module in one layer.

(a) Varying the number of layers.

| #layers | | 5 | 8 | 10 | 12 | 15 |
|---|---|---|---|---|---|---|
| w edge feature | Recovery(%) | 52.53 | 54.12 | 54 | **54.3** | 54.08 |
| | Perplexity | 4.3114 | 4.1986 | 4.1883 | **4.1536** | 4.2185 |
| w/o edge feature | Recovery (%) | 52.68 | 53.78 | 54.28 | 54.11 | **54.7** |
| | Perplexity | 4.3192 | 4.1983 | 4.1766 | 4.1829 | **4.1687** |

(b) Varying the number of vectors.

| #Vec. | 16 | 32 | 64 |
|---|---|---|---|
| Recovery (%) | 53.97 | **54.28** | 53.63 |
| Perplexity | **4.1598** | 4.1766 | 4.2636 |

(c) Varying the V-MLP.

| | Recovery (%) | Params |
|---|---|---|
| V-MLP | **54.28** | 4.2K |
| MLP | 53.42 | 36.0K |
| w/o V-MLP | 53.67 | **0.0K** |

(d) Varying the vector field.

| | Recovery (%) | Perplexity |
|---|---|---|
| Baseline | **54.28** | **4.1766** |
| w/o Vec. field | 35.43 | 7.5455 |
| w/o $g_{i,j}$ in Eq. equation 7 | 53.05 | 4.2907 |

(e) Varying the transformation $\mathcal{T}$.

| ID | Gbf | $\vec{\mathbf{h}}_k/\|\vec{\mathbf{h}}_k\|$ | Transformation | Recovery(%) | Perplexity |
|---|---|---|---|---|---|
| 1 | ✓ | ✓ | ✓ | **54.28** | **4.1766** |
| 2 | ✗ | ✓ | ✓ | 53.55 | 4.2495 |
| 3 | ✓ | ✗ | ✓ | 53.36 | 4.2238 |
| 4 | ✗ | ✗ | ✗ | 52.89 | 4.3268 |

in recovery scores, with the highest recovery achieved at 54.72% for 12 layers. Ablation experiments without the edge featurizer show that with or without edge features, the performance is comparable. Especially when the number of layers reaches 15, it even achieves better results, indirectly proving the effectiveness of VFN-IF and its potential for reducing reliance on hand-crafted features.

**The number of vectors.** In Table 6b we also perform an ablation study to determine the optimal number for vectors. It demonstrates that increasing the vector number from 16 to 32 brings a slight improvement in recovery score from 53.72% to 54.26%, while maintaining a low perplexity score of 4.14. However, further increasing the number to 64 results in a decrease in recovery score to 53.83% and a higher perplexity score of 4.20. Overall, these findings suggest that 32 is a more suitable choice.

**V-MLP.** In Table 6c, we observe that using V-MLP outperforms using a regular MLP or not using it at all. Compared to using a regular MLP, V-MLP significantly reduces the parameter count.

**Vector field design.** Results in Table 6d show that removing the whole vector field operator leads to a significant drop in recovery, indicating its importance in capturing protein folding patterns. The incorporation of the $g_{i,j}$ in the edge aggregation module also has a substantial effect on the performance of the model.

**Vector field operator design.** We validate the effectiveness of RBF and $\vec{\mathbf{h}}_k / \|\vec{\mathbf{h}}_k\|$ in Eq. equation 3, as shown in Table 6e. Furthermore, it illustrates that completely excluding the Eq. equation 3 and directly flattening the vectors $\mathbf{H}_{i,j}$ into $\mathbf{g}_{i,j}$ leads to an obvious performance decrease.

### A.5.1 VISUALIZATION RESULTS

In this section, we visualize some of the protein structure restoration results of VFN-IF, as shown in Figure 13, Figure 14 and Figure 12.

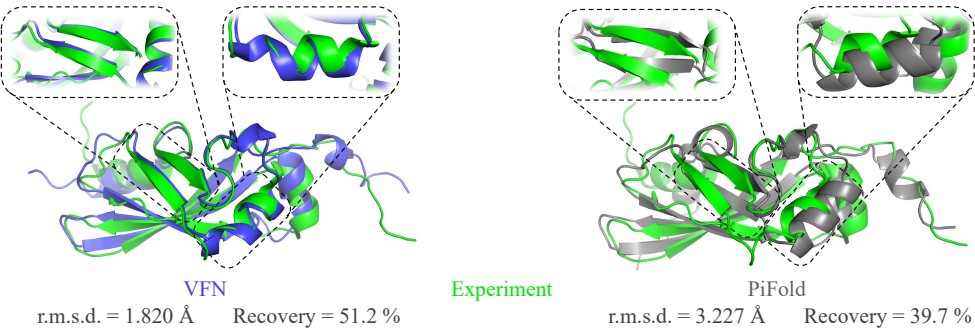



VFN       Experiment       PiFold

r.m.s.d. = 1.820 Å    Recovery = 51.2 %       r.m.s.d. = 3.227 Å    Recovery = 39.7 %



Figure 12: Visualization results of a challenging sample (PDB 2KRT). We use AlphaFold2 to recover the structure based on the predicted sequence and compare it against the experimentally determined ground-truth structure.

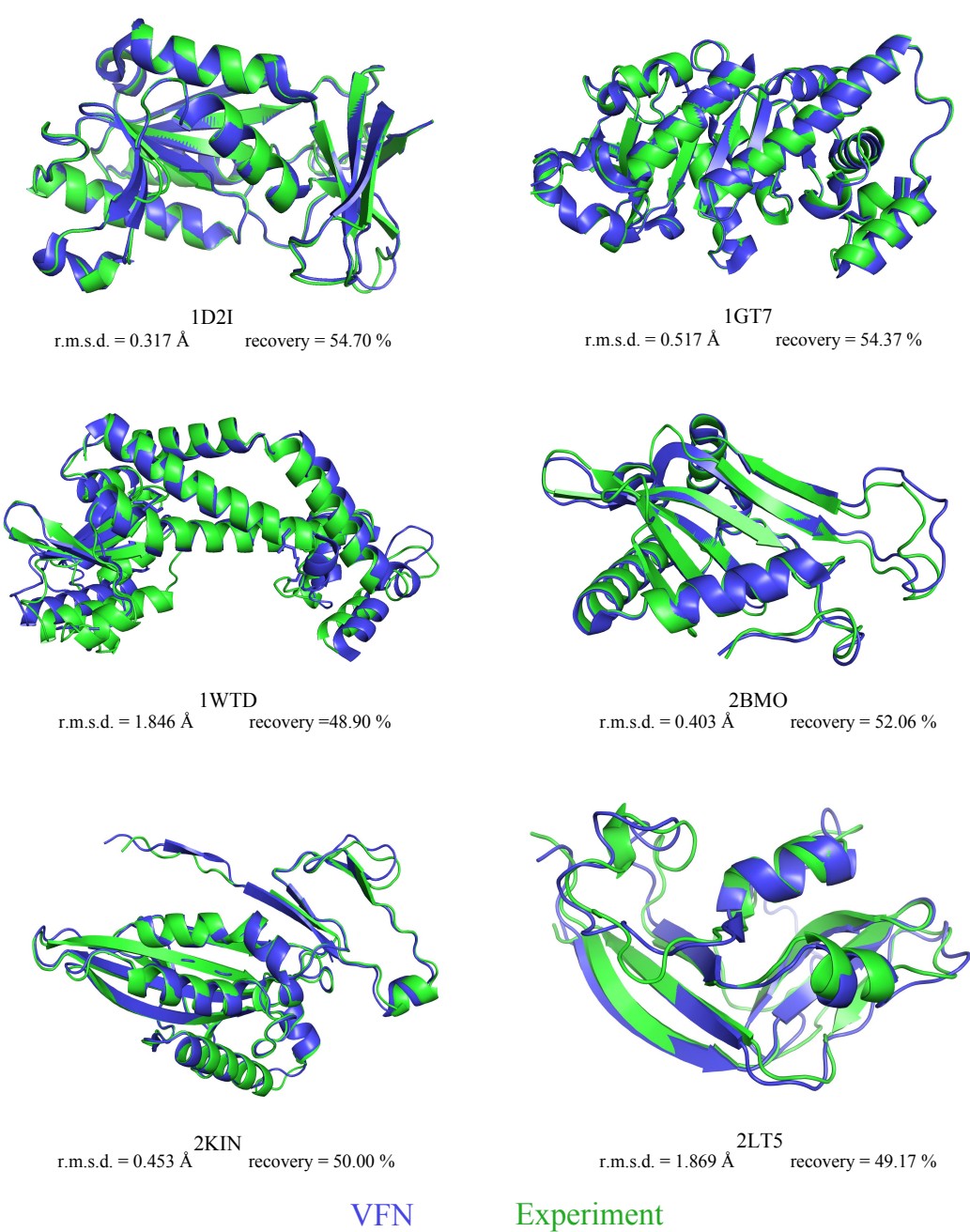

Figure 13: The visualization result of VFN.

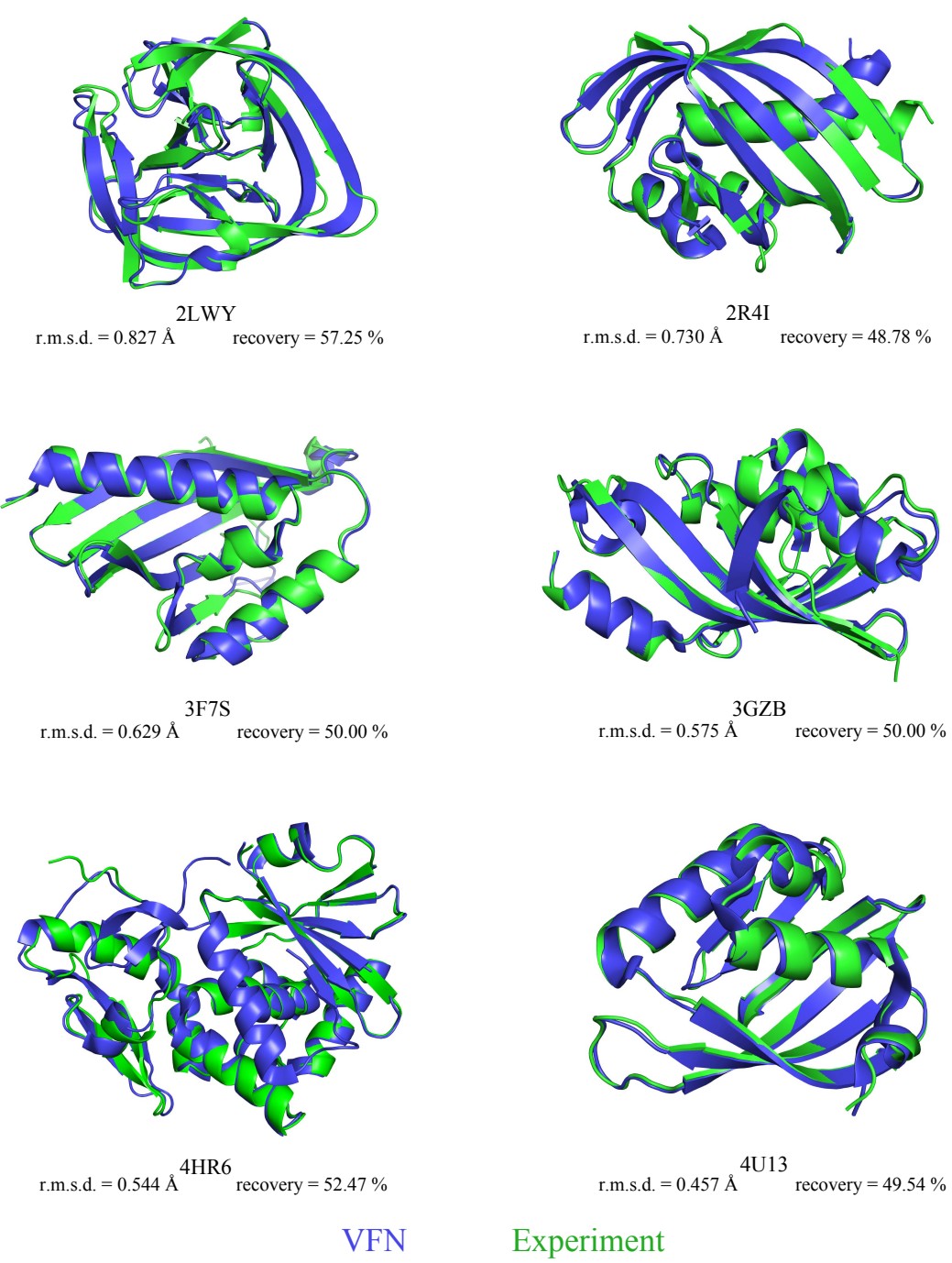

Figure 14: The visualization result of VFN.

