# OpenReview forum: "De novo Protein Design Using Geometric Vector Field Networks"
_ICLR.cc/2024/Conference — ICLR 2024 spotlight_

### Official Review · Reviewer_AKqf · 2023-10-24

**Soundness:** 3 good
**Presentation:** 2 fair
**Contribution:** 3 good
**Rating:** 8
**Confidence:** 4

**Summary:**

This paper presents a frame-based architecture for operating on protein structure. The key idea is to parameterize a set of virtual atoms within each residue frame. For each pair of interacting frames, the coordinates of both sets of virtual atoms in the destination frame are used to compute attention weights and message values. The aggregated message is then used to update the node features and virtual atom coordinates. In experiments, the authors swap out the architectures of PiFold and FrameDiff and achieve significantly improved performance under identical experimental settings.

**Strengths:**

* The paper establishes a new entry in the design space of residue frame-based architectures, an exciting direction for protein representation learning.
* The experimental results are quite strong and establish that VFN could be used as a drop-in replacement for alternative SOTA architectures.
* The non-exchangeable treatment of virtual atoms (unlike IPA) leaves open the possibility of using the framework for real sidechain atoms.

**Weaknesses:**

* The thesis of the paper would be improved by better contextualization relative to IPA. The authors should not shy away from acknowledging significant similarities, but highlight the key changes and the insights behind them. I would suggest a side-by-side algorithmic comparison.
* The paper could be further strengthened by additional comparisons with IPA. Particularly, if we replace IPA in AlphaFold/ESMFold with VFN, does the performance persist? It should not be too hard to run this experiment since the structural module is not very large.
* The claims about a "universal encoder" are not well-supported. It would be nice to see actual experiments where sidechains are also involved.
* From the novelty standpoint, an argument can be made that the architecture is similar enough to IPA and / or PiFold to count against its technical significance.

Justification for score: I think this is a good paper and am happy to recommend acceptance if the Questions are fully addressed.

**Questions:**

* Please describe more details on how the VFN-IF+ training split is constructed.
* Table 5 shows results for FrameDiff+ProteinMPNN vs VFN-Diff+VFN-IFE. What was the exact setting for reporting these numbers reported here, relative to Table 4? Are there equivalent results for FrameDiff+VFN-IF or VFN-Diff+VFN-IF? Please show these for all experimental settings in Table 4.
* In Figure 1, why does FrameDiff suffer in scTM more on medium-length proteins than the longest proteins?
* Why was it necessary to retrain FrameDiff, as claimed in the appendix?
* Please clarify if the PiFold and FrameDiff numbers are taken directly from the respective papers. Please affirm that the results and claims made here are indeed under *identical experimental conditions* relative to PiFold and FrameDiff, or if they have been modified, please be very direct about these modifications.

---

> ### Author Response · Authors · 2023-11-16
> **Part.1**
>
> Dear Reviewer:
>
> We thank the reviewer for the time and effort. The concerns raised by the reviewer are highly professional and meaningful. In the following, we will address each issue in detail and provide the necessary experimental results.
>
>
>
> ### 1) The issues regarding the presentation
>
> We appreciate the reviewer's constructive suggestions. Indeed, our current paper may not be reader-friendly for those unfamiliar with IPA. We will carefully address this issue and make substantial improvements to our writing in the next version. In the **global response**, we presented a comparison between VFN and IPA, and highlighted insights that were overlooked by the reviewer. Please refer to the global response for more details.
>
> ### 2) Replacing the IPA with VFN in AlphaFold2
>
> This is a very meaningful experiment that we have attempted before. However, training AlphaFold2 is resource-intensive, requiring approximately **2 weeks** of training with **128 TPUs**, as demonstrated in the official paper. Moreover, intricate data processing methods are employed during this process. Currently, we do not have sufficient resources to conduct this experiment, even for fine-tuning AlphaFold2. Therefore, we leave this experiment for future work. But thanks for the suggestion.
>
> ### 3) The comparative experiments on tasks related to sidechains.
>
> We think that experiments related to sidechains are also meaningful, so we conducted comparative experiments on the benchmark established within GearNet[1], as shown in the table below:
>
> | |GearNet[1]|VFN|
> |:-|:-:|:-:|
> |Fold3D (Acc) | 52.17 | **55.98** |
> |EC ($f_{max}$) | 76.22 | **77.74** |
> |GO-BP ($f_{max}$) | 44.05 | **44.96** |
>
> The above experimental settings were meticulously aligned, constituting rigorous ablation experiments. Due to time constraints, the training duration for both methods was set at 2/3 of the official setting. For detailed information on the specific benchmark, please refer to the GearNet paper. One advantage of VFN is that it **eliminates** the need for a **featurizer** to provide **hand-crafted features** such as atomic distances and directions. Instead, it simply requires **treating real atoms as virtual atoms** and feeding them into the vector field operator. This experimental outcome highlights **the potential** of our approach in future tasks related to **sidechain**.
>
> [1] Protein Representation Learning by Geometric Structure Pretraining
>
> ### 4) The issues regarding the novelty
>
> We believe that some insights regarding SE(3) invariance have been overlooked, leading to concerns related to novelty. Therefore, we elaborate on our insights in detail in the global response. We would greatly appreciate it if the reviewer could carefully read our global response to understand our insights. Our novelty and contributions mainly include the following points: 1) insights into **the atom representation bottleneck** in IPA, 2) **a novel method ensuring SE(3)-invariance**, and 3) **the vector field operator** (as depicted in equation 1-4). The VFN employs many **commonly used** operations, such as MLP-based attention (used in PiFold). However, these operations are **not the focus** of this paper, and the use of common operations should not undermine the novelty of our work.
>
> ### 5) The question regarding the the VFN-IF+ training split
>
> We initially **filtered out** proteins that exhibit a high degree of similarity (>**80%**) to sequences in the **validation set** and **test set**. Therefore, our training split does not pose any concerns regarding data leakage. Subsequently, proteins with a resolution greater than 9A were also excluded. Ultimately, our training set consists of 120,934 proteins.
>
> ### 6) The question regarding the scTM result on medium-length proteins
>
> We currently believe that this phenomenon is attributed to **data bias** in the PDB. Since this phenomenon is observed not only in FrameDiff but also in VFN and other published methods, the most likely cause is the bias in the training data. It is noteworthy that this phenomenon is particularly pronounced in scTM (Figure 1.B), while it is not evident in scRMSD (Figure 5). VFN also experiences this issue; however, VFN is more **robust** to data bias, and the problem is not as pronounced.
>
> **`Due to word limit, more information is shown on the next page`**

---

> ### Author Response · Authors · 2023-11-16
> **Part.2**
>
> ### 7) The question regarding the fairness of the comparative experiment.
>
> We understand the reviewer's concerns regarding the fairness of our experimental setting. However, we would like to clarify that **all comparative experiments in the main text** were conducted with not only **strict alignment** but also **fairness** in the experimental setup, except for VFN-IF+ which utilized a larger training dataset.
>
> ### 8) The question regarding the table 5
>
> The settings for Table 5 are as follows: Noise Scale = 1, Num. step = 500, Num Seq. = 1. We do have results for VFN-Diff+VFN-IF, as shown below:
>
> |Methods|scTM>0.5|RMSD<2|Diversity|
> |:-| :-:|-:|-:|
> |FrameDiff+ProteinMPNN |37.25%|6.88%|38.02%|
> |VFN-Diff+VFN-IF |44.81%|11.48%|42.72%|
> |VFN-Diff+VFN-IFE |**45.43**%|**11.73**%|**43.46**%|
>
>
> **We are conducting experiments** on more settings as per the reviewer's suggestions. However, the runtime for these experiments is exceedingly long. We plan to release these results soon. However, the lack of uniformity in the experimental settings within this field poses challenges for such experiments. The reasons are as follows:
>
> 1) The ProteinMPNN in Table 5 utilized a **larger training set**, **additional data augmentation**, and an autoregressive decoder, which is **unfair to VFN-IF**. We are incorporating these tricks employed by ProteinMPNN into VFN-IF to obtain experimental results.
>
> 2) The experimental results in Table 5 indicate that, even in an unfair setting with **fewer training data** and **no data augmentation**, VFN can still **outperform** the FrameDiff pipeline.
>
> 3) The ProteinMPNN in **Table 1** is reproduced by PiFold, validated by peer review, and **strictly aligned with our setting**. This experiment demonstrates that VFN-IF can significantly outperform Protein MPNN in a fair setting (**54.74%** vs. 45.96%).
>
> 4) VFN-IF employs PiFold's one-shot decoder, leading to faster inference speed. However, it is not easy to sample multiple results, so the Num Seq. can only be set to 1.
>
> 5) One fact is that, in this experiment, inverse folding is currently **not** the primary bottleneck and does not result in significantly noticeable differences. The performance of this table is determined by the diffusion model. The final results will be very similar with Table 4.
>
> ### 9) The question regarding the FrameDiff retraining
>
> FrameDiff is trained on the PDB. On one hand, the data in the PDB may increase daily, and on the other hand, the author of FrameDiff does not release the sample IDs used in their training. **If we do not retrain FrameDiff, it will result in misalignment of the training data.** To avoid this issue, we have retrained FrameDiff to ensure that the training sets for all comparative experiments are consistent.
>
> ### 10) The question regarding the identical experimental conditions
>
> Our comparative experiments with FrameDiff are **rigorously aligned without any tricks**. We directly adopted and reported the official results of PiFold, and our comparative experiments are also precisely aligned with PiFold. However, it is important to note that PiFold employs a featurizer, whereas VFN-IF **does not use a featurizer**. This constitutes an advantage for VFN, as PiFold relies on manually extracted distances and directions between atoms, which VFN does not require.
>
> ### 11) Emphasizing the Contributions of VFN-IFE
>
> The challenge of enhancing the performance of inverse folding using ESM remains unresolved. We propose a novel fine-tuning method to address this challenge. Our approach, VFN-IFE, significantly outperforms the ICML **Oral** paper, LM-Design[2] (**62.67%** vs. 55.65%).
>
> [2] Structure-informed Language Models Are Protein Designers. ICML 2023.

---

> ### Comment · Reviewer_AKqf · 2023-11-20
>
> The authors have addressed some of my concerns; however:
>
> (1) The contextualization relative to IPA should be more substantive and I would prefer to see this as a section in the Appendix at the very least. The authors should not hesitate to say that VFN can be viewed as building on IPA. The current manuscript does not mention IPA anywhere in the Methods section, which I believe is a misleading omission. It would strengthen, rather than weaken the paper to do a side-by-side comparison with IPA, highlighting the differences.
>
> (2) The new sidechain experimental results are poorly contextualized, and from a quick skim of the names of these tasks, it is not obvious to me that sidechain information is essential to these tasks.
>
> (3) The authors have revealed that the VFN-IFE+ training split is constructed with an 80% sequence similarity cutoff, which strikes me as far too high. It is not even clear whether sequence-based splits are appropriate; note that CATH is a structure-based split.
>
> (4) As for Table 5, even if ProteinMPNN is trained on more data, I'd like to see a head-to-head comparison VFN vs ProteinMPNN on a common set of generated backbones. In any case, it is shouldn't be hard to retrain ProteinMPNN if the authors have managed to retrain FrameDiff.
>
> Altogether, the response has not resolved most of my concerns and I will keep the current score.

---

> ### Author Response · Authors · 2023-11-23
> **Part 1**
>
> Dear Reviewer:
>
> We sincerely appreciate the responsible feedback from the reviewer. At the same time, we apologize for our delayed response, because, over the past week, we have been urgently conducting experiments aligning VFN-IF with ProteinMPNN, as mentioned by the reviewer. We have achieved astonishing results, with **VFN-IF outperforming the official ProteinMPNN by up to 14% in terms of scRMSD!!!** Additionally, we have addressed concerns regarding the comparison between IPA and VFN side by side. Therefore, we kindly request the reviewer to read the materials provided here. *We sincerely appreciate the reviewer's time and effort once again*. For better readability, we have uploaded a PDF as supplementary material. **Please download the supplementary material** (on openreview), which includes clearer experimental tables and a more **vivid side-by-side comparison**. Next, we will provide a detailed explanation for each of the questions you raised.
>
>
>
> ### **1) The Comparison between VFN and IPA**
>
> We agree with the reviewer and have incorporated the **updates to the paper** in **Appendix A.1** as per the reviewer's instructions. To facilitate the reviewer's examination, we have included this section in **Section 2 of the supplementary materials** we provided. **Please download the supplementary materials**, or review section A.1 in the updated paper for a clearer explanation. Here, for the sake of conciseness, we are only presenting crucial information beyond supplementary materials.
>
> 1) We vividly illustrate the comparison between IPA and VFN using pseudocode, as shown in Algorithm 1 and 2 in the supplementary material. This comparison is presented **side by side**. Due to the limitations of Markdown, we are unable to provide such vivid comparison here; please refer to our supplementary material.
> 2) From the intuitive comparison of pseudocode, VFN and IPA appear to be vastly different.
> 3) The reason we use IPA as an analogy for VFN in the paper is to facilitate better understanding for our readers. However, this does not imply a high degree of similarity between VFN and IPA. While VFN and IPA do share the same paradigm (virtual atoms), their specific methods are significantly different, as illustrated in the pseudocode.
> 4) We have revised and updated the paper as per the reviewer's instructions. In the next version, we will further enhance our presentation to better compare VFN and IPA.
>
> ### **2) Sidechain-related Experiments**
>
> Here, we begin by referencing the introduction to the relevant tasks in GearNet.
>
> > GearNet: Enzyme Commission (**EC**) number prediction seeks to predict the EC numbers of different proteins, which describe their catalysis of biochemical reactions. The EC numbers are selected from the third and fourth levels of the EC tree, forming 538 binary classification tasks. Gene Ontology (**GO**) term prediction aims to predict whether a protein belongs to some GO terms. Fold classification (**Fold3D**) is first proposed in Hou et al. (2018), with the goal to predict the fold class label given a protein.
>
> Those tasks are **full-atom**, and the majority are associated with protein functionality. Therefore, they constitute experiments related to side-chain interactions. In addition, many papers, e.g. [2], have been published based on the same tasks to investigate better representation methods related to side chains. The representation method of VFN is different from theirs, representing a potential future approach that can be integrated with existing methods. We leave this aspect of the research for future work.
>
> The experimental results are shown below:
> | |GearNet[1]|VFN|
> |:-|:-:|:-:|
> |Fold3D (Acc) | 52.17 | **55.98** |
> |EC ($f_{max}$) | 76.22 | **77.74** |
> |GO-BP ($f_{max}$) | 44.05 | **44.96** |
>
> [1] Protein Representation Learning by Geometric Structure Pretraining
> [2] Learning Hierarchical Protein Representations via Complete 3D Graph Networks
>
> ### **3) The issues regarding VFN-IFE+**
>
> This is indeed a significant issue. Thank you for the valuable suggestions, reviewer. We have removed VFN-IF+ from our Table 1. We apologize for this issue and have removed the relevant results in the updated paper (now available for download). However, it's important to note that the primary results of our work, namely VFN-IF, VFN-Diff, and VFN-IFE, remain unaffected, and this does not impact the core contributions of our paper.
>
> **`Due to word limit, more information is shown on the next page`**

---

> ### Author Response · Authors · 2023-11-23
> **Part 2: Important Experimental Results**
>
> ### **4) The issues regarding VFN-Diff + VFN-IF (`Important`)**
>
> Over the past week, we urgently aligned the settings of VFN-IF and ProteinMPNN and conducted experiments across the entire pipeline. The experimental results demonstrate that **VFN-IF outperforms the official ProteinMPNN by up to 14% in terms of scRMSD!!!** To the best of our knowledge, VFN-IF is **the new SoTA in designability (scRMSD and scTM) surpassing ProteinMPNN.**  Detailed experimental tables are presented more clearly in **Section 1 of the supplementary materials**; **Please download the supplementary material**. The information is repeated here for your convenience.
>
> `Note: In the end of this response, we also present the experimental results comparing RFDiffusion + ProteinMPNN with RFDiffusion + VFN-IF.`
>
> Noise Scale=0.1, Num. step=100, Num Seq.=8
> | |FrameDiff + ProteinMPNN |VFN-Diff + ProteinMPNN|VFN-Diff + VFN-IF|
> |:-|:-:|:-:|:-:|
> |$\text{scTM}_{0.5}$ | 76.67% | 83.83% | **90.49**% |
> |$\text{scRMSD}_{2}$ | 26.42% | 40.25% | **51.36**% |
>
> Noise Scale=0.1, Num. step=500, Num Seq.=8
> | |FrameDiff + ProteinMPNN |VFN-Diff + ProteinMPNN|VFN-Diff + VFN-IF|
> |:-|:-:|:-:|:-:|
> |$\text{scTM}_{0.5}$ | 77.41% | 83.95% | **93.46**% |
> |$\text{scRMSD}_{2}$ | 28.02% | 44.20% | **`58.27%`** |
>
> Noise Scale=0.5, Num. step=500, Num Seq.=8
> | |FrameDiff + ProteinMPNN |VFN-Diff + ProteinMPNN|VFN-Diff + VFN-IF|
> |:-|:-:|:-:|:-:|
> |$\text{scTM}_{0.5}$ | 76.42% | 81.23% | **91.60**% |
> |$\text{scRMSD}_{2}$ | 23.46% | 40.00% | **53.33**% |
>
> Noise Scale=1.0, Num. step=500, Num Seq.=8
> | |FrameDiff + ProteinMPNN |VFN-Diff + ProteinMPNN|VFN-Diff + VFN-IF|
> |:-|:-:|:-:|:-:|
> |$\text{scTM}_{0.5}$ | 53.58% | 67.04% | **72.84**% |
> |$\text{scRMSD}_{2}$ | 10.62% | 25.93% | **26.79**% |
>
> Due to the time limit during the discussion period, the program for the setting with num sequence = 100 is still running. This is because under this setting, each sample requires time-consuming ESMFold to predict 100 structures. We will include the results for this setting in the next version. However, **the metrics for num sequence = 8 are the official setting used by ProteinMPNN**, and the above experiments are sufficient to demonstrate the superiority of VFN-IF.
>
> *Diversity and novelty metrics are used to evaluate the diffusion model, not the performance of inverse folding.* For the sake of brevity, we have omitted diversity and novelty metrics from the table. If reviewer is interested, these metrics are detailed in Table 4 and Table 5 in the paper.
>
> ### **5) Enhancing RFDiffusion Performance with VFN-IF. (`Important`)**
>
> We replaced ProteinMPNN in the RFDiffusion (A concurrent study) pipeline with VFN-IF and achieved an also approximately **6% improvement** in designability. The results are as follows:
>
> |Methods|scTM>0.5|RMSD<2|
> |:-| :-:|-:|
> |RFDiffusion+ProteinMPNN |87%|35%|
> |RFDiffusion+VFN|**92**%|**41**%|
>
> Due to time constraints of discussion, the results above are based on the analysis of a protein with a length of 300 residues. However, this is sufficient to demonstrate the efficacy of VFN-IF.
>
> [1] De novo design of protein structure and function with RFdiffusion. Nature 2023.
>
> ### **5) The Issues Regarding the Additional Experiments**
>
> We will incorporate these additional experiments into our paper and release the source code for public use.

---

### Official Review · Reviewer_MXeu · 2023-10-31

**Soundness:** 3 good
**Presentation:** 2 fair
**Contribution:** 3 good
**Rating:** 6
**Confidence:** 5

**Summary:**

This paper proposes vector field network to model to better model local frames. Building on this VFN, this paper constructs two sequential models, respectively targeting at designing protein structures and generating protein sequence based on given protein backbone structure.  The paper achieves a new SOTA score on CATH 4.2 on inverse folding task and performs better than FrameDiff on protein structure design.

**Strengths:**

The proposed model achieves a new SOTA score on the CATH 4.2 benchmark.

**Weaknesses:**

1. **The architecture design lacks some novelty:** It seems for the protein structure design part (VFN-Diff) borrows some ideas from FrameDiff, while for the inverse folding part (VFN-IF), the virtual atom is similar to that of PiFold and the node interaction (Equation 4, 5, 6) is similar to the node gating mechanism in PiFold.

2. **The problem setting is unfair:** In the first paragraph of section 4, the author mentioned "In the protein diffusion part, the protein structure is designed and represented using backbone frames T. Subsequently, these backbone frames are fed into the inverse folding network to obtain the corresponding protein sequence for the designed structure." If I didn't understand wrongly, this means the author first design the protein structure  and then generate protein sequence based on the designed structures.  Therefore, it's kind of like a pipeline. However, in Table 1, for the sequence design task, the author only compared with the inverse folding models, say sequence design based on real structure instead of designed structures. I think a more fair comparison should compare to baseline like structure design model plus inverse folding model, such as FrameDiff + ProteinMPNN, RFDiffusion [1]+ProteinMPNN and also some structure-sequence co-design model like ProtSeed [2].

[1] De novo design of protein structure and function with RFdiffusion. Nature 2023.

[2] Protein Sequence and Structure Co-Design with Equivariant Translation. ICLR 2023.

3.**Lack of baselines:** The paper lacks some important baselines. For example, [2] for sequence part. For structure design part, the author only compares with FrameDiff, while the current SOTA protein structure design model is RFdiffusion [1]. Also, there is some other protein structure design model like SCMDiff [3],

[3] DIFFUSION PROBABILISTIC MODELING OF PROTEIN BACKBONES IN 3D FOR THE MOTIF-SCAFFOLDING PROBLEM. ICLR 2023.

[4] Generating Novel, Designable, and Diverse Protein Structures by Equivariantly Diffusing Oriented Residue Clouds. ICML 2023.

4. **The writing is unclear:** The author may need to use a unified annotation system, like sometimes {i, j} sometimes k is confusing while they mean the same thing. Additionally, the author may need to provide an overall graph of the architecture to help reader understand this paper.

**Questions:**

1. In figure 2, $T_{i\rightarrow j}$ means T_j to T_i, while in line below Equation 1, it means T_i to T_j. What does this term really mean?

2. Why he range of H can be negative, say -200 A?

3. $d_q$ is the number of channels in $g_{i,j}$. Does that mean the number of virtual atoms between node i and node j? What is the specific value used in this paper?

4. Are the MLP in equation 7 and 4 are the same one?

5. Using the ESM as initialization and then testing the model on CATH benchmark may have data leakage issues. How did the author deal with this problem?

---

> ### Author Response · Authors · 2023-11-16
> **Part.1**
>
> Dear Reviewer:
>
> We thank the reviewer for the time and effort. The concerns raised by the reviewer are highly professional and meaningful. However, there may be a **misunderstanding on the objective of our paper**, leading to those concerns raised. Please allow us to first provide an explanation of the objective of our paper before addressing each of the reviewer's concerns.
>
> ### 1) The objective of our paper is the protein encoder, not the design method.
>
> **VFN does not aim to propose a new protein design method, but rather introduces a novel protein structure encoder to enhance and replace the widely used encoder in protein design, IPA.** Improving IPA is highly meaningful, as acknowledged by the other two reviewers, **unsL** and **AKqf**. At the same time, our fair comparison experiments also demonstrate that VFN can indeed **significantly outperform IPA**. We understand that the concerns raised by the reviewer are *crucial* and *professional* *if our paper were about protein design methods*. However, this is a paper concerning the protein encoder specifically. We would sincerely appreciate it if the reviewer could consider our paper from **the perspective of protein encoder**.
>
> ### 2) The issues regarding the lack of novelty
>
> The VFN employs many **commonly used** operations, such as MLP-based attention (used in PiFold). However, these operations are **not the focus of this paper**, and the use of common operations **should not undermine** the novelty of our work. We have extensively elucidated our novelty and insights in the **global responses**. Please refer to global responses for more details. In short, **the core innovations** in this article revolve around: 1) insights into **the atom representation bottleneck** in IPA, 2) **a novel method ensuring SE(3) invariance**, and 3) **the vector field operator** (as depicted in equation 1-4). Regarding the virtual atoms in PiFold, the crucial point to emphasize is that **there are no virtual atoms within PiFold's GNN**. Its virtual atoms **only** exist in the featurizer, outside the GNN. This is distinctly different from the implicit representation of virtual atoms in VFN and IPA. Please avoid confusing these concepts.
>
>
> ### 3)  The issues regarding the inverse folding benchmarks and diverse settings
>
> We did provide a comparison of the settings (**VFN-Diff+VFN-IFE vs. FrameDiff + ProteinMPNN**), mentioned by the reviewer, in the appendix. Our method exhibits a **11.18% advantage** in designability. The experimental results are presented in the table below, and for details, please refer to **appendix** A.2 as well as Table 5 and Figure 10.
>
> |Methods|scTM>0.5|RMSD<2|Diversity|
> |:-| :-:|-:|-:|
> |FrameDiff+ProteinMPNN |37.25%|6.88%|38.02%|
> |VFN-Diff+VFN-IF |44.81%|11.48%|42.72%|
> |VFN-Diff+VFN-IFE |**45.43%**|**11.73%**|**43.46%**|
>
> In addition, we would like to emphasize the following:
> 1) Once again, **the purpose of VFN is not to propose a new protein design pipeline but rather to introduce a new protein design encoder.**
> 2) **The significance of the inverse folding benchmark we adopted.** Inverse folding is a crucial task, and the benchmark we adopted is widely recognized. At least **20** papers have conducted tests using the same or similar benchmarks on **real protein structures**, including **science** paper, ProteinMPNN, and ProtSeed mentioned by the reviewer (VFN-IF significantly outperforms ProtSeed, **54.7% vs. 43.8%**). Moreover, inverse folding can be widely used in various protein design pipelines, **not just protein diffusion**. The outstanding performance of VFN-IF is a **significant objective contribution**.
> 3) We partially agree with the reviewer's thoughts on benchmarks based on generated proteins. However, **such benchmarks face some obvious issues**. One fact is that current protein generation models still encounter many issues. Testing based on generated data introduces many **incorrect structures** and **biases** from the **generation model**. Additionally, the random seed of the generation model introduces random factors. Therefore, for inverse folding, such benchmarks may not be rigorous enough. The CATH benchmark adopted by the community may be better, so we follow the CATH benchmark.
>
> **`Due to word limit, more information is shown on the next page`**

---

> ### Author Response · Authors · 2023-11-16
> **Part.2**
>
> ### 4) The issues regarding the lack of baselines
>
> Thank you for the reviewer's valuable suggestions. In the following, **we present a comparison between VFN-Diff and the methods mentioned by the reviewer**. The experimental results demonstrate the significant **superiority of VFN-Diff**. However, before showcasing the results, we would like to reiterate that the focus of our paper lies in proposing **a new encoder** rather than a designing method. The comparison between VFN and FrameDiff is sufficient to validate the effectiveness of our approach.
> ##### 4.1) **Comparison with SCMDiff.**
> VFN-Diff significantly surpasses SCMDiff[1]. The comparative results are presented in the following table, with strict alignment of experimental settings.
>
> |Methods|scTM>0.5|RMSD<2|
> |:-|:-:|-:|
> |VFN-Diff|**67.04%**|**25.93%**|
> |SCMDiff|1.92%|1.92%|
>
> [1] DIFFUSION PROBABILISTIC MODELING OF PROTEIN BACKBONES IN 3D FOR THE MOTIF-SCAFFOLDING PROBLEM. ICLR 2023.
>
> ##### 4.2) **Comparison with Genie.**
> VFN-Diff significantly surpasses Genie[2]. The comparative results are presented in the following table, with strict alignment of experimental settings (the best setting for Genie). Since Genie is **unable** to generate proteins exceeding a length of **256**, the comparison provided below is limited to proteins with a length less than 256.
>
> |Methods|scTM>0.5|RMSD<2|Diversity|
> |:-|:-:|-:|-:|
> |VFN-Diff|**89.38%**|**50.00%**|**77.50%**|
> |Genie|73.75%|25.00%|72.50%|
>
> [2] Generating Novel, Designable, and Diverse Protein Structures by Equivariantly Diffusing Oriented Residue Clouds. ICML 2023.
>
> ##### 4.3) **Comparison with RFDiffusion.**
> When both VFN and RFDiffusion do not utilize pretraining weights, VFN can surpass RFDiffusion and achieve SoTA. The comparisons of designability (**median scRMSD, lower values are better**) are presented in the table below:
>
> |Protein Length |70|100|200|300|
> |:-| :-:|:-:|:-:|:-:|
> |RFDiffusion |12.5|14.5|20.2|26.0|
> |VFN-Diff |**0.8**|**0.9**|**2.1**|**2.2**|
>
> In an unfair setting (RFDiffusion w/ pretraining, VFN w/o pretraining), **VFN stands out as the method closest to RFDiffusion**:
>
> | |RFDiffusion|VFN-Diff|FrameDiff|
> |:-|:-:|:-:|:-:|
> |scRMSD<2 |**100%**|89%|35%|
>
> As highlighted in the FrameDiff paper, RFDiffusion employs pretraining trick, **more training data and neural network parameters**. Directly comparing with RFDiffusion is unfair. While VFN-Diff could certainly enhance performance using the same approach (e.g. more training data), this is beyond the scope of our focus, and the comparison of RFDiffusion should not be a weakness of our paper. We will add those results into our paper.
>
> **`Due to word limit, more information is shown on the next page`**

---

> ### Author Response · Authors · 2023-11-16
> **Part.3**
>
> ### 5)  The issues regarding the writing
>
> We appreciate the reviewer's constructive feedback. We acknowledge that there are some issues with the writing in our paper. We will heed your suggestions and make improvements to the presentation of our paper in the next version.
>
> ### 6)  The question regarding $\mathbf{T}_{i\leftarrow j}$
>
> The reviewer is correct. There is a typo here. We confirm that $\mathbf{T}_{i\leftarrow j}$ signifies the transition from $\mathbf{T}_j$ to $\mathbf{T}_i$.
>
> ### 7)  The question regarding $\vec{\mathbf{h}}_k$
>
> As $\vec{\mathbf{h}}_k$ is a vector, it possesses a direction. The positive or negative sign of $\vec{\mathbf{h}}_k$ signifies the direction of the vector.
>
> ### 8)  The question regarding $d_q$
> Sorry, there is a typo here. The number of channels for $\mathbf{g}_{i,j}$ is $d_g$, not $d_q$; please refer to Eq.3. The reviewer's understanding of $d_q$ is correct, referring to the quantity of atoms. We used 16 atoms in the paper, which is much less than the quantity in IPA.
>
> ### 9)  The question regarding Eq.7 and Eq.4
>
> Those MLPs are **not** the same one. The two MLPs do not share parameters. We will emphasize this point in the next version.
>
> ### 10)  The question regarding ESM model
>
> The use of ESM to assist inverse folding[3] has been **accepted by peer review** (**LM-design[3], ICML 2023 Oral**). We adopted exactly the same setting. Therefore, we believe this is **not a flaw** in our paper. While we acknowledge the risk of data leakage, we argue that it should be considered from a different perspective.
>
> 1) The objective of inverse folding is to map structures into sequences. Since ESM has not been trained on structures but only on sequences, this does not constitute a data leak in the strict sense.
>
> 2) The approach[3][4] of inverse folding based on ESM has demonstrated its efficacy in sequence design. Therefore, from this standpoint, it holds significance.
>
> 3) Even from the strictest perspective, LM-Design and VFN-IFE can be considered as retrieval-based sequence design methods. However, in the ESM training set, potentially thousands of sequences exhibit a high degree of similarity to the predictions of VFN-IF. **Retrieving** the correct sequences remains an ongoing challenge, and significant progress has been made by VFN-IFE in addressing this issue (**62.67%** vs. 55.65%, compared to LM-design[3]).
>
> [3] Structure-informed Language Models Are Protein Designers. ICML 2023.
> [4] Knowledge-Design: Pushing the Limit of Protein Deign via Knowledge Refinement

---

> > ### Comment · Reviewer_MXeu · 2023-11-17
> > **Reply to the rebuttal**
> >
> > I appreciate the authors' efforts in providing empirical evidence to demonstrate the effectiveness of their method. I have two follow-up questions:
> >
> > 1. **Goal of this paper:** Can the authors further elaborate the meaning of protein encoder? The authors keep emphasizing their model is a protein encoder in the responses. Did that mean the goal of this paper is to learn better protein representation? Then the evaluation benchmark should not be limited to design tasks only, but also should include some understanding tasks like protein binding affinity prediction.
> >
> > 2. **Data leakage in ESM:** For the data leakage problem of ESM, the author mentioned LM-design accepted by ICML 2023. However, I also discussed this issue with the authors of LM-design before and they said they actually used some data filtering techniques to prevent data leakage issue. The authors said you followed exactly the same setting as LM-design. Can the author explain the data filtering techniques you used in this paper?

---

> > > ### Author Response · Authors · 2023-11-21
> > > **Part 2: Response to the data leakage in ESM.**
> > >
> > > ### **4) Alignment of VFN Setting with LM-Design: A Comprehensive Explanation**
> > >
> > > We directly utilized **the data provided by the official LM-Design GitHub repository**, including training data, validation data, and test data. This dataset is entirely **consistent with the data reported in the LM-Design paper**, as indicated in the following quotation. If LM-Design employed any data filtering methods, these would also be applied to our dataset, given that we utilized the data provided by LM-Design.
> > >
> > > > Description of the Dataset in LM-Design[8]: We mainly compared LM-DESIGN against recent strong baselines on the CATH 4.2 dataset, using the same data splits as the compared systems, e.g., Structured Transformer, GVP, ProteinMPNN, and PiFold, where proteins were partitioned by the CATH 4.2 topology classification, resulting in **18024** proteins for training, **608** proteins for validation, and **1120** proteins for testing.
> > >
> > > The above description can be found in the supplementary material B.1 section of LM-Design. The data we utilized aligns entirely with the official information provided by LM-Design and corresponds precisely to the descriptions in the paper.
> > >
> > > [8] Structure-informed Language Models Are Protein Designers. ICML 2023 Oral.
> > > ### **5) LM-Design Employs Data Filtering for Additional Data, While VFN Does Not Utilize This Additional Data.**
> > >
> > > LM-Design[8] has only one description for data filtering, aimed at avoiding data leakage. However, this data filtering operation is designed to mitigate data leakage caused by **additional training data**. In contrast, VFN-IFE **does not utilize such additional training data**, rendering the need for these data filtering operations unnecessary.
> > >
> > > Specifically, LM-Design introduces an additional setting, in which it employs AlphaFold2's prediction results as supplementary training data. Some of these training data exhibit high similarity to the CATH dataset, necessitating the application of filtering. In contrast, VFN does not incorporate these training data, eliminating the need for such data filtering methods. The relevant descriptions in LM-Design pertaining to this aspect are as follows:
> > >
> > > > LM-Design: LM-DESIGN works well with data augmentation (Hsu et al., 2022) via incorporating predicted structures from AlphaFold 2. We perform different scales of data augmentation, the details of data processing are described in **Appendix E**.
> > >
> > > > **Appendix E** of LM-Design: In order to prevent data leakage introduced by data augmentation, we need to exclude proteins that have the same fold as the proteins in validation and test splits. ...
> > >
> > > [8] Structure-informed Language Models Are Protein Designers. ICML 2023 Oral.

---

> > > > ### Comment · Reviewer_MXeu · 2023-11-22
> > > > **Reply to the rebuttal**
> > > >
> > > > Thanks for the further clarification and experiments! Most of my concerns are addressed by the authors. Therefore, I raised my score from 5 to 6. I think the author needs to explain the motivation and goal of this paper more clearly in the revised version. Besides, the author also should add the necessary experimental results which have been done during rebuttal process into the paper to help reader to better understand this paper.

---

> > > > > ### Author Response · Authors · 2023-11-23
> > > > > **The final important information, check it please.**
> > > > >
> > > > > Dear Reviewer:
> > > > >
> > > > > We sincerely appreciate the reviewer for adjusting the scores in response to our feedback, and we are committed to enhancing the quality of our paper writing. However, we have a **crucial update** to share that we believe will **further raise the scores** upon the reviewer's consideration. **Kindly allow us 5 minutes for this important information.**
> > > > >
> > > > > We found that VFN-IF is **the first method (new SoTA) capable of surpassing the designability of ProteinMPNN (a paper published in 'Science')** and **achieving up to a 14% improvement in scRSMD**!!! Additionally, we observed that VFN-IF **can further enhance the designability of RFDiffusion by 6%**. This implies that VFN not only contributes to protein representation but also **holds significance in protein design methods!** This finding addresses concerns raised by the reviewer in the first round of feedback.
> > > > >
> > > > > ### **1) The Updated Experimental Results of VFN-IF.**
> > > > > > Reviewer: ... such as FrameDiff + ProteinMPNN, RFDiffusion [1]+ProteinMPNN ...
> > > > >
> > > > > Because you and reviewer AKqf both raised concerns regarding VFN-Diff+VFN-IF, we urgently aligned the settings of ProteinMPNN and VFN-IF in the past week. The experimental results are provided in **Section 1** of the supplementary material. **Please download the supplementary material** (on openreview), which includes clearer experimental tables. We compared the performance of VFN-IF and ProteinMPNN in generating protein structures, with VFN demonstrating outstanding performance. We repeat some parts of findings here.
> > > > >
> > > > > Noise Scale=0.1, Num. step=500, Num Seq.=8
> > > > > | |FrameDiff + ProteinMPNN |VFN-Diff + ProteinMPNN|VFN-Diff + VFN-IF|
> > > > > |:-|:-:|:-:|:-:|
> > > > > |$\text{scTM}_{0.5}$ | 77.41% | 83.95% | **93.46**% |
> > > > > |$\text{scRMSD}_{2}$ | 28.02% | 44.20% | **`58.27%`** |
> > > > >
> > > > > Noise Scale=0.5, Num. step=500, Num Seq.=8
> > > > > | |FrameDiff + ProteinMPNN |VFN-Diff + ProteinMPNN|VFN-Diff + VFN-IF|
> > > > > |:-|:-:|:-:|:-:|
> > > > > |$\text{scTM}_{0.5}$ | 76.42% | 81.23% | **91.60**% |
> > > > > |$\text{scRMSD}_{2}$ | 23.46% | 40.00% | **53.33**% |
> > > > >
> > > > > ### **2) Enhancing RFDiffusion Performance with VFN-IF.**
> > > > >
> > > > > We replaced ProteinMPNN in the RFDiffusion pipeline with VFN-IF and achieved an approximately **6% improvement** in designability. The results are as follows:
> > > > >
> > > > > |Methods|scTM>0.5|RMSD<2|
> > > > > |:-| :-:|-:|
> > > > > |RFDiffusion+ProteinMPNN |87%|35%|
> > > > > |RFDiffusion+VFN-IF|**92**%|**41**%|
> > > > >
> > > > > Due to time constraints, the results above are based on the analysis of a protein with a length of 300 residues. However, this is sufficient to demonstrate the efficacy of VFN-IF.
> > > > >
> > > > > ### **3) The Issues Regarding Novelty**
> > > > >
> > > > > > Reviewer: It seems for the protein structure design part (VFN-Diff) borrows...
> > > > >
> > > > > VFN is a highly innovative approach. In Section 2 of the supplementary materials, we clearly demonstrate the differences between VFN and IPA through pseudocode. The reason we use IPA as an analogy for VFN in the paper is to facilitate better understanding for our readers. However, in reality, IPA and VFN are entirely different in terms of methodology and implementation. We hope this clarification can address the concerns raised by the reviewer.
> > > > >
> > > > >
> > > > >
> > > > > ### **4) The Issues Regarding Presentation**
> > > > >
> > > > > We appreciate your suggestions regarding the paper presentation. Indeed, we have addressed some typos pointed out by the reviewer, such as issues between "i" and "j," and have updated our paper accordingly. The revised version is now available for download. In the next iteration, we will further enhance the paper based on the reviewer's feedback.
> > > > >
> > > > > ### **5) The Issues Regarding the Additional Experiments**
> > > > >
> > > > > We will incorporate these additional experiments into our paper and release the source code for public use.

---

> ### Author Response · Authors · 2023-11-21
> **Part 1: Response for the goal of this paper**
>
> Dear Reviewer:
>
> Thank you for your response and thorough review of our paper. Below, we address your concerns in detail and provide **empirical evidence (experiments)** supporting the effectiveness of VFN.
>
> ### **1) VFN's Objective: Residue Frame Representation, Commonly Utilized in Protein Design**
>
> > Reviewer: Can the authors further elaborate the meaning of protein encoder? The authors keep emphasizing their model is a protein encoder in the responses. Did that mean the goal of this paper is to learn better protein representation?
>
> Yes, the goal of VFN is to improve **residue frame representation**, a common type of protein representation.  Protein representation encompasses two common forms of representation: **atom-based** representation and **residue frame-based** representation. The distinction of frame representation is that the most of atoms within proteins are typically **unknown**, and each amino acid is treated as a rigid body (a residue frame). It is a fact that the frame representation paradigm[1,2,3,4,5] is **frequently utilized in protein design tasks** because, in protein design tasks, atoms are often unknown, and frame representation is a better choice. Therefore, VFN, as a protein encoder **tailored for frame representation**, conducts its main experiments based on **protein design**, which is a reasonable choice. The significance of frame representation has also been acknowledged by the other two reviewers, **unsL** and **AKqf**.
>
> [1] Protein Sequence and Structure Co-Design with Equivariant Translation. ICLR 2023.
> [2] De novo design of protein structure and function with RFdiffusion. Nature 2023.
> [3] SE(3) diffusion model with application to protein backbone generation. ICML 2023.
> [4] Protein structure generation via folding diffusion.
> [5] Generating Novel, Designable, and Diverse Protein Structures by Equivariantly Diffusing Oriented Residue Clouds. ICML 2023.
>
> ### **2) VFN Targets in Residue Frame Representation – Distinct From Conventional Protein Understanding Tasks**
> > Reviewer: Then the evaluation benchmark should not be limited to design tasks only, but also should include some understanding tasks like protein binding affinity prediction.
>
> The protein understanding task [6,7,8] typically employs atom-based representation, **not residue frame representation**. This is because, in protein understanding tasks, atoms are usually known, and there is no need to use a residue frame representation designed for atom-unknown scenarios. VFN, on the other hand, is a GNN designed **for residue frame representation**. Therefore, the protein understanding task is not suitable for evaluating VFN.
>
> [6] Protein representation learning by geometric structure pretraining. ICLR 2023.
> [7] Unsupervised Protein-Ligand Binding Energy Prediction via Neural Euler's Rotation Equation.
> [8] EQUIBIND: Geometric Deep Learning for Drug Binding Structure Prediction. ICML 2022.
>
> ### **3) Experimental Support for VFN as the Pioneering Frame-Based GNN with Atomic Representation (Protein Understanding) Compatibility**
>
> > Reviewer: Then the evaluation benchmark should not be limited to design tasks only, but also should include some understanding tasks like protein binding affinity prediction.
>
> If we are not mistaken, VFN is the first frame-based GNN with compatibility for atom-based representation (protein understanding). Experimental evidence (protein understanding) supporting this claim will be presented in the following.
>
> | |GearNet[6]|VFN|
> |:-|:-:|:-:|
> |Fold3D (Acc) | 52.17 | **55.98** |
> |EC ($f_{max}$) | 76.22 | **77.74** |
> |GO-BP ($f_{max}$) | 44.05 | **44.96** |
>
> The above experiments utilized the benchmark proposed by GearNet[6]. Subsequently, we present an overview within GearNet of these tasks:
>
> > GearNet: Enzyme Commission (**EC**) number prediction seeks to predict the EC numbers of different proteins, which describe their catalysis of biochemical reactions. The EC numbers are selected from the third and fourth levels of the EC tree, forming 538 binary classification tasks. Gene Ontology (**GO**) term prediction aims to predict whether a protein belongs to some GO terms. Fold classification (**Fold3D**) is first proposed in Hou et al. (2018), with the goal to predict the fold class label given a protein.
>
> The above experimental settings were meticulously aligned, constituting rigorous ablation experiments. Due to time constraints, the training duration for both methods was set at 2/3 of the official setting.
>
> The experimental results demonstrate that VFN not only exhibits superior performance in frame representation **but also proves compatible with atom-based representation (protein understanding), showcasing promising potential in the domain of protein understanding.**
>
> [6] Protein representation learning by geometric structure pretraining. ICLR 2023.
>
> **`Due to word limit, more information is shown on the next page`**

---

### Official Review · Reviewer_unsL · 2023-10-31

**Soundness:** 3 good
**Presentation:** 3 good
**Contribution:** 2 fair
**Rating:** 8
**Confidence:** 4

**Summary:**

The authors introduce a new architecture (VFN) for processing protein structures, which allows to represent residue positions with many virtual atoms. This allows for more fine-grained modeling of residue interactions. The proposed VFN architecture is shown to outperform standard architectures in both protein generation using diffusion models and inverse folding tasks.

**Strengths:**

I agree with the authors that there has been an over-reliance on IPA in the literature for protein tasks. It makes a lot of sense to investigate improvements to it, so the paper does target a very important problem in my eyes.
Introducing effectively more data channels into the model to increase its capacity is also very sensible. Importantly the model is shown to improve the results on the most common and important protein modeling tasks.

**Weaknesses:**

The general reasoning of why the proposed architecture works better and should be constructed the way it is lies on the concept of atom representation bottleneck. But this bottleneck is not really introduced or investigated in a rigorous manner. Maybe the authors can at least give concrete theoretical counter examples of what problem can be modeled with VFN but not IPA. At least an experimental ablation on varying the virtual node count would be interesting to see how the performance changes.

It would also be nice if authors could show the theoretical expressivity of their proposed construction. E.g. is there something it cannot model or is it provably universal.

General GNNs have seen some improvements on modeling equivariant structures, e.g. [1, 2, 3] which in some ways have some similarities to the current work (e.g. higher dimensional embedding in frame averaging is a bit like virtual nodes here). It would be nice to see how the proposed VFN architecture stacks up to those general GNN constructions. Although admittedly, those papers usually test their models on molecular and other physics-inspired 3D tasks, but not proteins tackled in this paper. However the extension, especially in case of frame averaging or the multi-channel EGNN, would be trivial. Note that there are also many more improved 3D GNNs, especially in molecule domain that could be applied to this problem. I would like to see the authors test against at least a few of these options, especially frame-averaging as it has been used for proteins a couple of times now [4, 5] and would tackle a similar problems as the proposed model in a very general way.

Speaking of proteins, in the abstract authors say that only basic encoders such as IPA have been proposed for proteins so far. E.g. [4] applies frame averaging [2] to antibodies, with a quite intersting non-relational architecture for antibody design. While its restricted to a certain protein family it's still worth mentioning that 'less simple' encoders do exist, at least in specific cases.

[1] Du, Weitao, et al. "A new perspective on building efficient and expressive 3D equivariant graph neural networks."

[2] Puny, Omri, et al. "Frame averaging for invariant and equivariant network design."

[3] Levy, Daniel, et al. "Using Multiple Vector Channels Improves E (n)-Equivariant Graph Neural Networks."

[4] Martinkus, Karolis, et al. "Abdiffuser: Full-atom generation of in-vitro functioning antibodies."

[5] Jin, Wengong, et al. "Unsupervised Protein-Ligand Binding Energy Prediction via Neural Euler's Rotation Equation."

**Questions:**

I would mainly like the authors to provide a more detailed theoretical and/or experimental analysis of the introduced atom representation bottleneck, as I mentioned in the weaknesses.


-------
### After Rebuttal
Thank you for the extensive rebuttal. I read through all the reviews and all the answers and I think the work has noticeably improved.
It's a bit strange the authors have not updated the paper itself with all the new results, but I trust that they will for the final version.
I now recommend acceptance.

---

> ### Author Response · Authors · 2023-11-16
> **Part.1**
>
> Dear Reviewer:
>
> We thank the reviewer for the time and effort. In the following, we will address each issue in detail and provide the necessary experimental results.
>
> ### 1) The issues regarding the concept of atom representation bottleneck.
>
> > Reviewer: 'The general reasoning of why the proposed ...'
>
> In our **global response**, we rigorously defined the atom representation bottleneck and the underlying insight using formulas. Please refer to the global response for more details. In brief, the way IPA satisfies SE(3) invariance **restricts the use of activation functions**. Subsequently, we will address the specific questions raised by the reviewer. It is important to note that these responses will **utilize the notation established in the global response**.
>
> 1) **Theoretical counter example.** According to global responses, in IPA, 1) the **activation function** cannot be applied to the virtual atomic coordinates $\vec{\mathbf{q}}_l,\vec{\mathbf{k}}_l$, whereas VFN can. 2) The $h \in \mathbb{R}$ generated by IPA is a scalar and **cannot represent direction**, whereas the vector $\vec{\mathbf{h}}_k \in \mathbb{R}^3$ generated by VFN can. 3) In IPA, the operator for calculating $h$ **lacks learnable parameters** and cannot yield specific interatomic features (as illustrated in Figure 2.B of the paper), but VFN can.
>
> 2) **Ablation study on varying the virtual node count.** We are conducting ablation experiments on the diffusion model, but this typically takes 2-3 weeks for training. Therefore, we firstly present experiments based on the inverse folding here. Following the reviewer's guidance, we provide the following experiments concerning the virtual node count (the number of $\vec{\mathbf{h}}_k$). Upon completion of the experiments on the diffusion model, we will include them in the revised version.
>
> |Num. of $\vec{\mathbf{h}}_k$|4|8|16|32(Default)|
> |:-:|:-:|:-:|:-:|:-:|
> |Acc|53.08%|53.75%|54.12%|**54.28**%|
>
> ### 2) VFN possesses a unique advantage in theoretical expressivity.
> > Reviewer: 'It would also be nice if authors could show the theoretical expressivity ...'
>
> In the previous question, we actually addressed the differences between IPA and VFN in this regard, such as the **inability** of $h$ to represent **direction** and the **limitations in using activation functions**. This is already a solid fact. However, beyond this, we would like to emphasize **another advantage** of our approach. Protein modeling currently involves two tasks: **frame modeling** and **atomic modeling**. For frame modeling (diffusion), **PiFold cannot directly be compatible**, and for atomic modeling (inverse folding), **IPA also faces challenges**. In contrast, VFN can handle both tasks, and the reasons are as follows:
> 1) PiFold is not designed for frame modeling, very different.
> 2) VFN, due to the flexibility of the vector field operator, **can treat real atoms as virtual atoms**, thus allowing the extraction of features between real atoms. However, IPA **cannot** employ such an approach. Because IPA **lacks learnable parameters** when computing features between the atomic coordinates $\vec{\mathbf{q}}_l,\vec{\mathbf{k}}_l$. Clearly, using distance pooling cannot extract features between specific real atoms, as shown by the equation in the global response:
> $h=\sum_l \Vert\vec{\mathbf{q}}_l-\vec{\mathbf{k}}_l\Vert^2 \quad$.
>
> **`Due to word limit, more information is shown on the next page`**

---

> ### Author Response · Authors · 2023-11-16
> **Part.2**
>
> ### 3) The issues regarding other GNN, such as frame averaging.
>
> > Reviewer: 'General GNNs have seen some improvements on modeling equivariant structures ...'
>
> In this part, we incorporate these methods into our study, conducting experiments to compare and elucidate the distinctions between VFN and these approaches.
>
>
> **Experimental Results**: Once again, due to the very long training time required for the diffusion model, we present the results of inverse folding here, and the results on diffusion will be included in the next version. The following experimental settings are **rigorously aligned**. We implemented frame averaging and multi-channel EGNN very carefully, ensuring that they achieve their intended effects. However, despite these efforts, these methods still lag noticeably behind VFN.
>
> |Methods|VFN-IF|Frame Averaging|multi-channel EGNN|
> |:-:|:-:|:-:|:-:|
> |Acc|**54.70%**|47.33%|45.57%|
>
> **Discussion on frame averaging:** In comparison to frame averaging, the approach employed by VFN to maintain SE(3) invariance is **more concise**, while also introducing an **local inductive bias**. Specifically, each amino acid is treated as a rigid body, inherently possessing the properties of a local frame. VFN directly leverages the properties of local frames, circumventing the complicated virtual frame construction in frame averaging. Additionally, since VFN utilizes the local frame, it introduces a local inductive bias similar to CNN (Convolutional Neural Network). In contrast, frame averaging constructs frames based on global atoms, lacking this particular attribute.
>
> **Discussion on IPA:** In comparison to IPA, the advancement of VFN lies **not only** in higher-dimensional embedding. As mentioned in the global response, the more important factors of VFN are: 1) VFN can utilize **activation functions** 2) VFN introduces **learnable parameters** 3) The $\vec{\mathbf{h}}_k$ in VFN can represent **direction**.
>
> **Discussion on other works:** We have carefully examined the related works provided by the reviewer, and we confirm that VFN cannot be simply stacked on these methods. For instance, frame averaging proposes a method to maintain SE(3) invariance, and we have also introduced a method to preserve SE(3) invariance. However, these two methods are not compatible. Additionally, some of these works ([1][2][3]) are contemporaneous with our paper, so we do not delve into detailed discussions on those. However, we will cite these papers in the next version.
>
> [1] A new perspective on building efficient and expressive 3D equivariant graph neural networks.
> [2] Abdiffuser: Full-atom generation of in-vitro functioning antibodies.
> [3] Using Multiple Vector Channels Improves E (n)-Equivariant Graph Neural Networks.
>
> ### 4) The issues regarding numerous protein structure encoders
>
> > Reviewer: 'Speaking of proteins, in the abstract authors say that only...'
>
> Thank you for the thorough review. We will enhance the precision of our expression, as pointed out by the reviewer. It's essential to note that in IPA, each node is a frame represented by the origin coordinates and rotation matrix. In Abdiffuser, however, each node is represented by the atomic coordinates of amino acids. Therefore, the task in Abdiffuser is not entirely identical to the residue frames modeling in IPA. In our abstract, we are referring to IPA's frame modeling. We will enhance our clarity in the next version.
>
>
> ### 5) Emphasizing the Contributions of VFN-IFE
>
> The challenge of enhancing the performance of inverse folding using ESM remains unresolved. We propose a novel fine-tuning method to address this challenge. Our approach, VFN-IFE, significantly outperforms the ICML **Oral** paper, LM-Design[4] (**62.67%** vs. 55.65%).
>
> [4] Structure-informed Language Models Are Protein Designers. ICML 2023.

---

> ### Author Response · Authors · 2023-11-21
> **Anticipating your valuable feedback and the opportunity for further constructive discussion.**
>
> We sincerely appreciate the time and effort you dedicated to reviewing our paper. In our response (also the global response), we have conscientiously addressed the meaningful concerns and suggestions raised by the reviewer. Please don't hesitate to let us know if you have any further questions.

---

> ### Author Response · Authors · 2023-11-23
>
> Dear Reviewer:
>
> We have addressed the concerns raised by the reviewer in our initial response, and we look forward to further discussion with you. However, as the discussion is coming to a close, we can only provide a final update and emphasize some additional information. We hope this helps resolve your concerns. Once again, we appreciate the time and effort you have dedicated to reviewing our work.
>
> ### **1) The Comparison between VFN and IPA**
>
> > Reviewer: 'The general reasoning of why the proposed ...'
>
>
> In our initial response, we have already presented relevant information in the global response. Here, to further clarify the distinctions between VFN and IPA, we have provided a vivid comparison based on pseudocode in the supplementary materials. **Please download the supplementary material** (on openreview) for this comparison.
>
> VFN is a highly innovative approach. In Section 2 of the supplementary materials, we clearly demonstrate the differences between VFN and IPA. The reason we use IPA as an analogy for VFN in the paper is to facilitate better understanding for our readers. However, in reality, IPA and VFN are entirely different in terms of methodology and implementation, as shown in the pseudocode. We hope this clarification can address the concerns raised by the reviewer.
>
> ### **2) The Updated Experimental Results of VFN-IF.**
>
> Here, we present a new experimental update. The latest experimental results demonstrate that VFN-IF is **the new SoTA in designability (scRMSD and scTM) surpassing ProteinMPNN**. VFN-IF outperforms the official ProteinMPNN by up to **14% in terms of scRMSD!!!** In Section 1 of the supplementary materials, we have presented relevant experimental results. Please refer to Section 1 of the supplementary materials for clearer experimental tables. We repeat part of findings here.
>
> Noise Scale=0.1, Num. step=500, Num Seq.=8
> | |FrameDiff + ProteinMPNN |VFN-Diff + ProteinMPNN|VFN-Diff + VFN-IF|
> |:-|:-:|:-:|:-:|
> |$\text{scTM}_{0.5}$ | 77.41% | 83.95% | **93.46**% |
> |$\text{scRMSD}_{2}$ | 28.02% | 44.20% | **`58.27%`** |
>
> Noise Scale=0.5, Num. step=500, Num Seq.=8
> | |FrameDiff + ProteinMPNN |VFN-Diff + ProteinMPNN|VFN-Diff + VFN-IF|
> |:-|:-:|:-:|:-:|
> |$\text{scTM}_{0.5}$ | 76.42% | 81.23% | **91.60**% |
> |$\text{scRMSD}_{2}$ | 23.46% | 40.00% | **53.33**% |
>
> We hope that the comprehensive information we have provided can address your concerns.

---

### Author Response · Authors · 2023-11-16
**The Global Response Part.2**

### 5) The Insight and Novelty of VFN

The core insights and novelty of VFN can be summarized as follows: 1) insights into **the atom representation bottleneck** in IPA, 2) **a novel method ensuring SE(3)-invariance**, and 3) **the vector field operator** (as depicted in equation 1-4). The VFN employs many **commonly used** operations, such as MLP-based attention (used in PiFold). However, these operations are not the focus of this paper, and the use of common operations should not undermine the novelty of our work. We will carefully improve our presentation in the next version.

---

### Author Response · Authors · 2023-11-16
**The Global Response Part.1**

Dear Reviewer:

We would like to express our gratitude to all the reviewers for their time and effort. We address common issues raised by the reviewers in this **global response**.

The reviewers collectively raised questions about **the atom representation bottleneck** and **the vector field operator**, and we explain them in detail here.

### 1) What is the atom representation bottleneck
IPA is designed for modeling the residue frames. It launches virtual atoms under the residue frame to represent the positional information of the local frame. Subsequently, IPA models the positions of these atoms to extract the geometric features between two frames. This can be written as:
$h=\sum_l \Vert\vec{\mathbf{q}}_l-\vec{\mathbf{k}}_l\Vert^2 \quad$ (causing **the atom representation bottleneck**, explained later)
Here, $h \in \mathbb{R}$ represents the output geometric features and serves as an attention bias; $\textbraceleft \vec{\mathbf{q}}_l\in \mathbb{R}^3\textbraceright$ and $\textbraceleft \vec{\mathbf{k}}_l\in \mathbb{R}^3\textbraceright$ are the sets of atom coordinates for the $i$-th and $j$-th amino acid, respectively; $l \in \textbraceleft1,...,L\textbraceright$ denotes the atom index. Evidently, this operator is a summation pooling operation that can **only yield a scalar**, and **no learnable parameters are involved**, severely lacking in expressive capability. This scalar **fails to convey directional information** between atoms and **cannot provide learnable weights for each atom** (as depicted in Figure 2(B) of our paper). This limitation restricts the network's capacity, prompting us to term this issue as the atom representation bottleneck.

### 2) The bottleneck is the defect in satisfying SE(3)-invariance
In IPA, $\vec{\mathbf{q}}_l,\vec{\mathbf{k}}_l$ is with respect to **the global frame**. However, on the other hand, IPA requires this operator to **satisfy SE(3)-invariance**. Therefore, in order to meet SE(3)-invariance, only operations similar to distance pooling can be employed in this operator. Once operations resembling **activation functions** are appiled on  $\vec{\mathbf{q}}_l,\vec{\mathbf{k}}_l$, SE(3)-invariance cannot be maintained. We refer to this constraint as **the SE(3)-invariance limitation**.

### 3) A new SE(3)-invariance method without limitations on the operator

In order to satisfy SE(3)-invariance while removing the limitation on the operator, we **propose** an ingenious **frame-based** approach for this problem: Our method directly places both $\vec{\mathbf{q}}_l$ and $\vec{\mathbf{k}}_l$ under **the $i$-th residue frame** $\mathbf{T}_i$, not the global frame adopted by IPA, using equation 1 in our paper. Thus, the SE(3)-invariance is ensured by $\mathbf{T}_i$, as proved in supplementary (A.1.6). The advantage is that, unlike IPA, our method imposes **no additional constraints** on the operator, allowing for using various operations, including **activation functions** (activation functions is unavailable for $\vec{\mathbf{q}}_l, \vec{\mathbf{k}}_l$ **in IPA**).

### 4) VFN bypasses the bottleneck via a novel vector field operator

With the method to maintain SE(3)-invariance, we devised the vector field operator and a following MLP layer to bypass the atom representation bottleneck.
1) The vector field operator functions as a vector linear layer, extracting geometric feature vectors $\{\vec{\mathbf{h}}_k\}$, denoted as:
$\vec{\mathbf{h}}_k=\sum_l w^\text{a} _{kl} \vec{\mathbf{q}}_l + \sum_l w^\text{b} _{kl} \vec{\mathbf{k}}_l$
here $\vec{\mathbf{h}}_k \in \mathbb{R}^3$ is the $k$-th output channel; $w^\text{a} _{kl},w^\text{b} _{kl} \in \mathbb{R}$ are the learnable weights. The vector field operator can extract information that $h$ (computed by IPA) cannot represent. For instance, $\vec{\mathbf{h}}_k$ can convey **directional information**, which $h$ cannot capture. Additionally, the learnable weights, $w^\text{a} _{kl},w^\text{b} _{kl}$, can **provide independent weights** for each atom to extract features, as illustrated in Figure 2 of the paper. In contrast, IPA lacks learnable weights and cannot provide independent weights for each atom.
2) As there is no limitation on operators, we designed an MLP with **an activation function** to handle $\vec{\mathbf{h}}_k$, as shown in Equation 4 of the paper. The introduction of the activation function enhances the expressive power of VFN. This sets VFN apart from IPA, as IPA cannot apply activation functions to $\vec{\mathbf{q}}_l,\vec{\mathbf{k}}_l$.

**`Due to word limit, more information is shown on the next page`**

---

### Public Comment · ~Wanru_Zhuang1 · 2023-11-30
**Clarification Request on Virtual Atom Coordinates Initialization in Algorithm 2**

Dear Authors,

I hope this message finds you well. I have carefully read your paper focusing on the topic of protein inverse folding, which I find particularly fascinating.

I am writing to inquire about a specific detail in the paper that I believe is crucial for a comprehensive understanding of your methodology. My question pertains to Algorithm 2 in Appendix A.1.1, which describes the VFN module.

As stated in Section 4.2, "It’s important to note that the coordinates of backbone atoms are used to initialize some of the virtual atom coordinates." Nevertheless, upon my review, it seems that the explicit process for the initialization of these virtual atom coordinates within Algorithm 2 is not provided. I am concerned that I might have missed this detail, and I would greatly appreciate it if you could shed light on where exactly in the algorithm, or perhaps elsewhere in the paper, this initialization is outlined.

Your clarification on this matter will not only satisfy my curiosity but also enhance the understanding of the algorithm for fellow researchers who are following your valuable work.

Thank you for your time and consideration. I am looking forward to your response.

---

### Meta-Review · Area_Chair_yujn · 2023-12-05

**Metareview:**

This study highlights a key limitation in current SE3 equivariant architectures used for protein modeling: their lack of expressiveness in representing atoms within residues and their interactions between residues. In response, the authors introduce a novel equivariant model, the Vector Field Network (VFN), which effectively addresses this issue. VFN demonstrates superior performance over standard protein architecture models in tasks like protein generation and inverse folding. This work represents a significant advancement in a crucial area and is likely to gain widespread usage.

**Justification For Why Not Higher Score:**

Though concrete and useful, the modifications proposed are not entirely surprising. There is no theory supporting the claims. More baselines could have been added also.

**Justification For Why Not Lower Score:**

The proposed architecture makes sense and is novel. Th work is well written and addresses an important problem. The numerical results demonstrate a clear benefit.

---

### Decision · Program_Chairs · 2024-01-16

Accept (spotlight)